# Modelling the mass budget and future evolution of Tunabreen, central Spitsbergen

Johannes Oerlemans[1], Jack Kohler[2], Adrian Luckman[3]

[1]Institute for Marine and Atmospheric Research, Utrecht University, Princetonplein 5, Utrecht, 3585CC, The Netherlands
[2]Norsk Polarinstitutt, Hjalmar Johansengate 14, Trømso, Norway 9296
[3]Department of Geography, Swansea University, Singleton Park. Swansea, SA2 8PP, United Kingdom

*Correspondence to*: Johannes Oerlemans (j.oerlemans@uu.nl)

**Abstract**. The 26-km long tidewater glacier Tunabreen is the most frequently surging glacier in Svalbard, with four documented surges in the past hundred years. We model the evolution of this glacier with a Minimal Glacier Model (MGM), in which ice mechanics, calving and surging are parameterized. The model geometry consists of a flowband to which three tributaries supply mass. The calving rate is set to the mean observed value for the period 2012-2019, and kept constant. For the past 120 years, a smooth Equilibrium Line Altitude (ELA) history is reconstructed by finding the best possible match between observed and simulated glacier length. There is a modest correlation between this reconstructed ELA history and an ELA history based on meteorological observations from Longyearbyen.

Runs with and without surging show that the effect of surging on the long-term glacier evolution is limited. Due to the low surface slope and associated strong height-mass balance feedback, Tunabreen is very sensitive to changes in the ELA. For a constant future ELA equal to the reconstructed value for 2020, the glacier front will retreat by 8 km during the coming hundred years. For an increase of the ELA of 2 m per year, the retreat is projected to be 13 km and Tunabreen becomes a land-terminating glacier around 2100.

The calving parameter is an important quantity: increasing its value by 50 % has about the same effect as a 35 m increase in the ELA, with the corresponding equilibrium glacier length being 17.5 km (as compared to 25.8 km in the reference state). Response times vary from 150 to 400 years, depending on the forcing and on the state of the glacier (tidewater or land-terminating).

## 1 Introduction

Tunabreen is a 26-km long tidewater glacier in central Spitsbergen (Figure 1). It is the most frequently surging glacier in Svalbard, with four documented surges during the past hundred years (Flink et al., 2015; Luckman et al., 2015). The surges occurred during 1924-1930 (advance 3 km), 1966-1971 (advance 2.1 km), 2002-2004 (advance 2 km), and 2016-2018 (advance 1.1 km). It appears that the frequency of surging has been increasing, with shorter duration of the periods of advance (Figure 2). Analysis of crevasse patterns visible on high-resolution satellite images has shown that surging is initiated close to the glacier front and then propagates upward (Flink et al., 2015).

Svalbard has a wide spectrum of surging glaciers, with surge periods ranging from a couple of decades to (presumably) a few hundred years (Lefauconnier and Hagen, 1991). Many of the larger tidewater glaciers on Svalbard do surge regularly (a precise percentage is not known). Glaciers with well-documented surge behaviour in a similar size-class as Tunabreen are for instance Kongsvegen (length $\approx$ 25 km, slope $\approx$ 0.028), Comfortlessbreen (length $\approx$ 16 km, slope 0.042) and Monacobreen (length $\approx$ 40

km, slope ≈ 0.027). The characteristic slope of Tunabreen (≈ 0.032) is within the usual range of larger tidewater glaciers. The observed long-term (multi-annual) calving rate is relatively high (≈ 270 m a$^{-1}$), but it should be noted that calving rates vary enormously across the Archipelago (Blaszczyk et al., 2009).  The surges of Tunabreen appear to be initiated on the lowest part of the glacier and propagate upward, which is not uncommon for surging tidewater glaciers in Svalbard (Sevestre et al., 2018). Altogether, Tunabreen appears to be a fairly representative glacier, albeit with a relatively high and increasing surge frequency.

Apart from the length fluctuations related to the surges, over the past hundred years Tunabreen has become shorter by about 1.5 km. This is likely due to an increasing Equilibrium Line Altitude (ELA) caused by rising air temperature (Førland et al., 2011), perhaps in combination with larger calving rates associated with higher ocean temperature (Luckman et al., 2015). Before 1920 the lower parts of Tunabreen and of Von Postbreen were joined to form one front in the Tempelfjorden. Length data back to 1870 exist (Flink et al., 2015), but are not considered here for model validation because the focus is on the period that Tunabreen was an independently calving glacier.

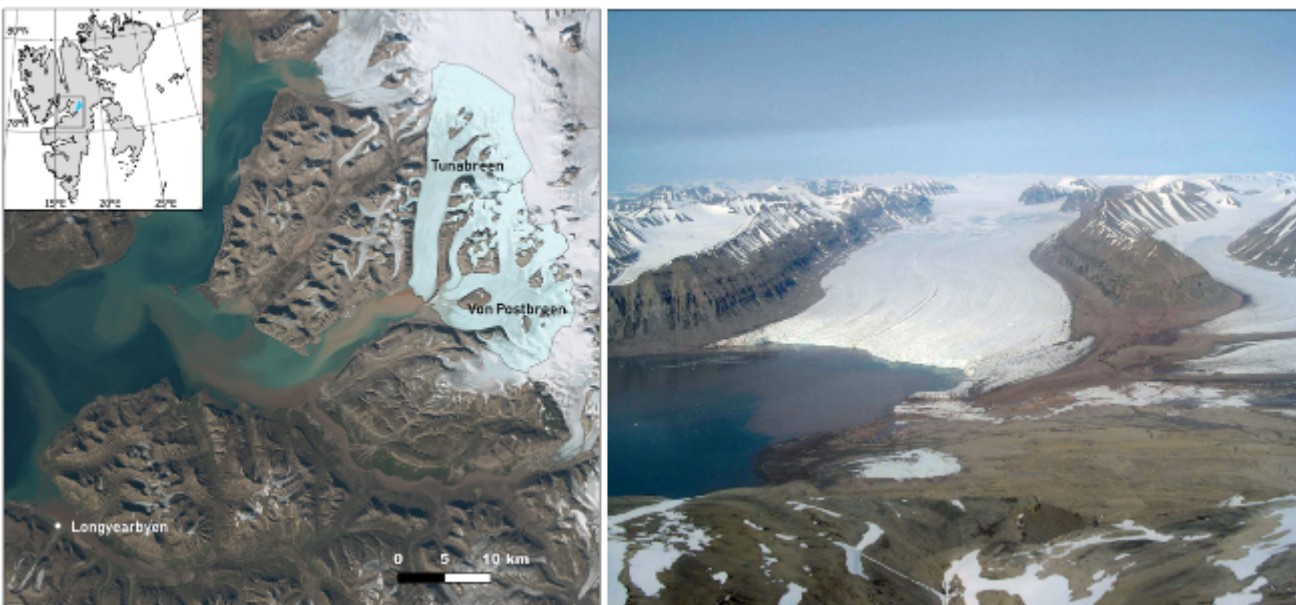

**Figure 1. Left: Location of Tunabreen and von Postbreen in Svalbard (inset). Background image is from a 2020 Sentinel-2 mosaic (https://toposvalbard.npolar.no). Right: Photograph of Tunabreen in 2015 (© Anders Skoglund, Norwegian Polar Institute).**

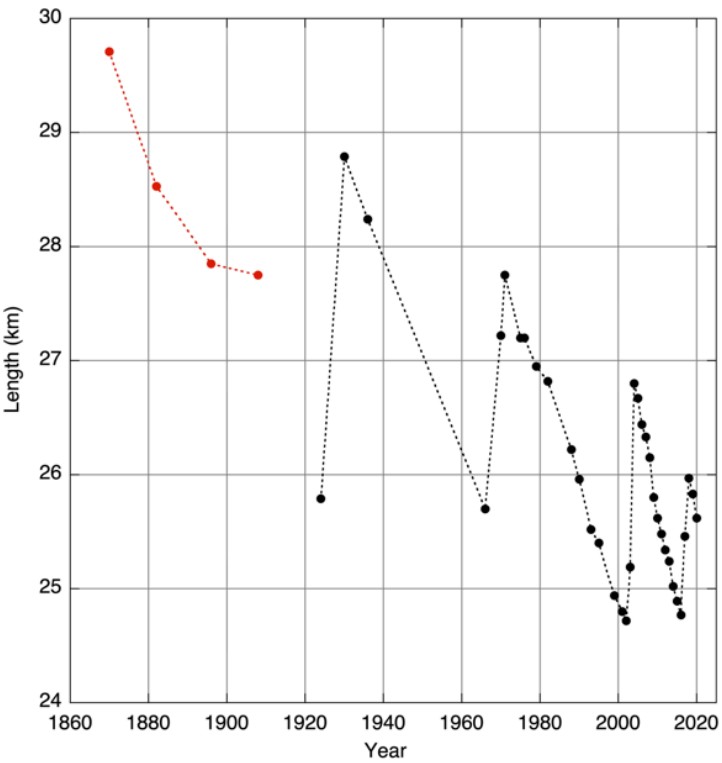

**Figure 2. Length of Tunabreen from historical documents, aerial photographs and satellite images (Flink et al. 2015, with additions). The first four data points in red refer to the period that Tunabreen and Von Postbreen formed a joined front.**

The goal of this study is to analyse the mass budget of Tunabreen, to determine how sensitive the glacier is to ongoing and future climate warming, and to see if the frequent surging has an effect on its long-term retreat.

The model we used is a so-called Minimal Glacier Model (MGM, Oerlemans 2011), in which dynamic processes are parameterized. In this class of models, ice mechanics are not explicitly considered, but a number of essential feedback mechanisms can be dealt with (height - mass balance feedback, effect of reversed bed slopes, variable calving rates, effect of

regular surging on the long-term mass budget, inclusion of tributary glaciers and basins). The basic idea behind MGMs is that, with respect to long-term evolution of glaciers, ice mechanics are 'slaved' by the exchange of mass with the environment (atmosphere and ocean). However, the most important mechanical effect, namely that glaciers are thicker when they are longer and/or rest on a bed with a smaller slope, should always be accounted for. We note that the model version employed here is similar to the one used in a study of Monacobreen in northern Spitsbergen (Oerlemans, 2018).

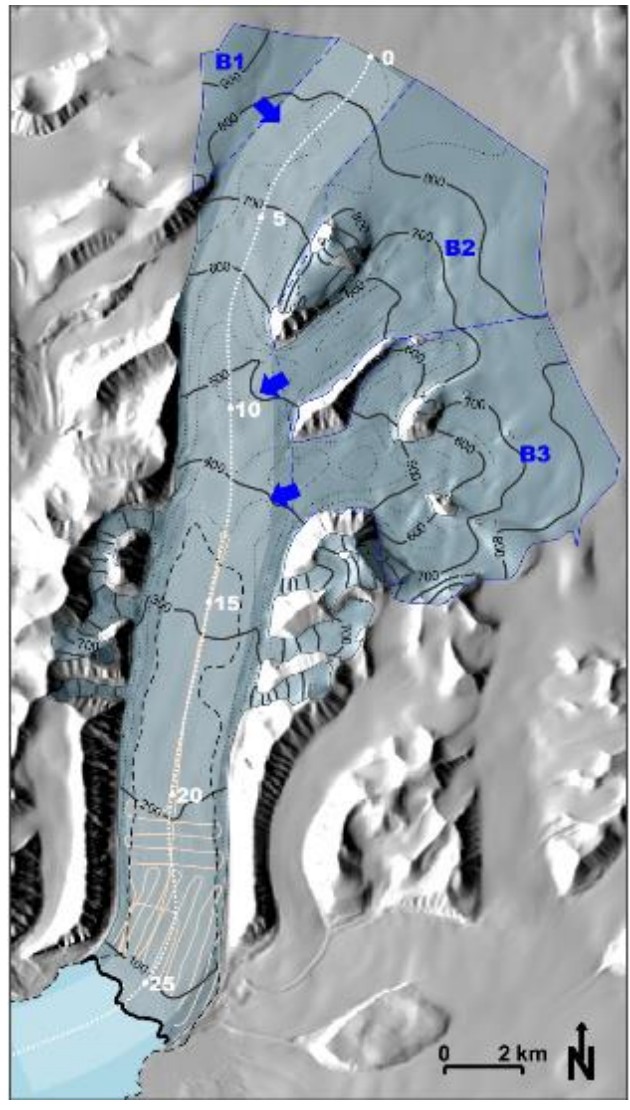

**Figure 3. Map of Tunabreen. Surface contours (thick black lines) at 100-m intervals are from the NPI S0 DEM (Norwegian Polar Institute, 2014), based on aerial photography from 2009 (lower glacier tongue, 0-450 masl) and 2011 (upper glacier, > 450 masl). Bed contours (dotted black lines) at 100-m intervals are based on the Fürst et al (2018) reconstruction in the upper glacier, and 2015 helicopter radar (K. Lindback & J. Kohler, in Welty et al 2020) measurements (light-red lines) in the lower glacier. Dashed subglacial contour shows the part of the bed below sea level. Center line profile (dotted white line) along 2200-m wide main flowband (lighter blue) shows distance from the ice divide at 5-km intervals. Three upper basins (blue dashed outlines) supply mass to the main flowband (blue arrows).**

 To our knowledge a quantitative model has never been constructed for Tunabreen. We consider our study as a first step, with the focus on the integrated mass-budget dynamics rather than ice mechanical processes. It is obvious that MGMs cannot simulate more subtle processes like the seasonal cycle of the calving flux, or the effect of high-water input (melt or rain) on sliding velocities. However, we believe that MGMs are able to provide first-order estimates of the relation between glacier size and climatic regime.

**2 Glacier model**

Tunabreen is modelled as a stream (flowband) of length $L$ and constant width $W$. Three tributary glaciers / basins supply mass to the main stream if they have a positive mass budget (Figure 3). A few minor tributaries are neglected because they have no significant effect on the total mass budget of Tunabreen. The $x$-axis, originating at the ice divide, follows the centre line of the flowband. $L$ is measured along this axis. Defined in this way, the glacier stand in 2009 serves as a reference point, with $L = 25.8$ km. The average width of the flowband is taken as 2200 m.

Conservation of mass (or volume, since ice density is considered to be constant) determines the evolution of the glacier. It can be formulated as:

$$\frac{dV}{dt} = F + M_m + \sum_{i=1}^{3} M_i = M_{tot} .$$  (1)

Here $V$ is the volume of the main stream of Tunabreen, $F$ is the volumetric calving flux ($< 0$), $M$ is the volumetric surface mass budget of the main stream, and the $M_i$ are the contributions from the tributary glaciers. In the following sections a number of parameterizations are introduced concerning the global ice mechanics, geometry, calving, and climate forcing.

We stress that a Minimal Glacier Model is fundamentally different from another class of approximate models that have become popular, so called lumped-parameter models (e.g. Fowler et al., 2001; Benn et al. 2019). In the MGM the basic idea is to have a model description that deals with the conservation laws integrated over an entire glacier. This is essential if one wants to compute the evolution of a glacier for imposed environmental change. In the lumped-parameter model the focus is more on the details of local mechanical processes and their interaction with hydrology and thermodynamics - the large-scale glacier parameters are then specified.

## 2.1 Prognostic equation for glacier length

The glacier volume $V$ (of the main stream) is given by $WL\bar{H}$ , where $\bar{H}$ is the mean ice thickness. Differentiating with respect to time yields

$$\frac{dV}{dt} = W \frac{d}{dt}(L\bar{H}) = W \left( \bar{H} \frac{dL}{dt} + L \frac{d\bar{H}}{dt} \right) = M_{tot} .$$  (2)

The mean ice thickness is parameterized as (Oerlemans, 2011)

$$\bar{H} = S(t) \frac{\alpha}{1+v\bar{s}} L^{1/2}.$$  (3)

Here $\bar{s}$ is the mean bed slope over the glacier length, and thus varies in time when the glacier length changes. $S$ is the "surge function", making it possible to impose a surge (to be discussed later). For a non-surging glacier we simply have $S(t) = 1$. The parameter $\alpha$ can be interpreted a measure of the global basal resistance of the bed to ice flow and determines the mean ice thickness for a given slope and glacier length. Because of the abundant presence of soft and presumably mostly saturated sediments at the bed, the larger glaciers on Svalbard experience rather low resistance and therefore are comparatively thin (e.g. Benn and Evans, 2010). As shown later, the value of $\alpha$ is therefore relatively low (typically $\sim 2$ m$^{1/2}$ as compared to a value of $\sim 3$ m$^{1/2}$ for many mid-latitude glaciers, or even $\sim 3.5$ m$^{1/2}$ for glaciers that are (partly) cold-based, like McCall glacier in Alaska (Oerlemans, 2011). The parameter $v$ determines the dependence of the mean ice thickness on the mean bed slope. Based on a large number of numerical experiments with an ice flow model (Oerlemans, 2001), its dimensionless value was set to 10.

Substituting the time derivative of Eq. (3) into Eq. (2) gives

$$\frac{dV}{dt} = \frac{W\bar{H}}{S} \left\{ L \frac{dS}{dt} + \frac{3}{2} S \frac{dL}{dt} \right\}$$  (4)

Rearranging then yields the prognostic equation for $L$:

$$125 \quad \frac{dL}{dt} = \frac{2}{3} \frac{M_{tot}}{W\bar{H}} - \frac{2}{3} \frac{1}{S} \frac{dS}{dt} L \qquad (5)$$

We have neglected a term involving $\frac{\partial \bar{s}}{\partial L}$ because it is generally small. However, the dependence of the mean ice thickness on the mean bed slope is still retained through Eq. (3). See section 5 in Oerlemans (2011) for more discussion.

The next step is to determine the mass budget, and we first consider the surface budget. The balance rate is assumed to be a linear function of altitude relative to the equilibrium-line altitude $E$ (see the compilation in Oerlemans and Van Pelt, 2015):

$$130 \quad \dot{b} = \beta(h - E) , \qquad (6)$$

Where $\beta$ is the balance gradient, $h$ is surface elevation and $E$ is the equilibrium-line altitude.

The total mass loss or gain of the flowband is found by integrating the balance rate over the glacier length:

$$B_s = \beta W \int_0^L [H(x) + b(x) - E] dx = \beta W (\bar{H} + \bar{b} - E) L , \qquad (7)$$

where $\bar{b}$ is the mean bed elevation of the glacier (note that this quantity as well as $\bar{H}$ depends on the glacier length, which in 135 fact introduces the height - mass balance feedback).

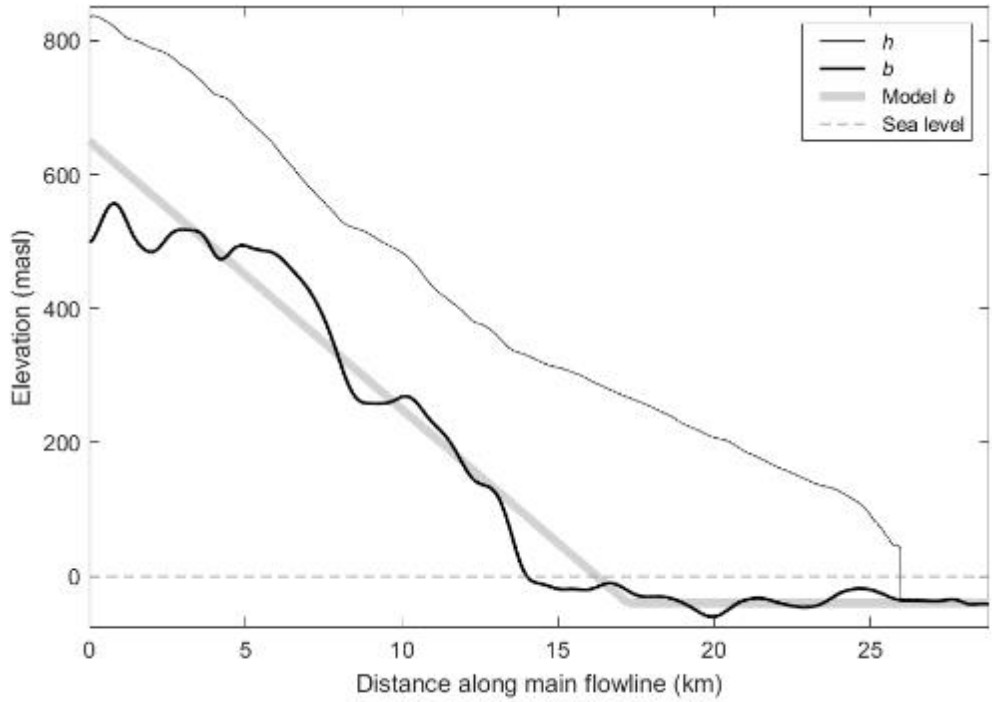

**Figure 4. Cross-section along centerline profile (shown in Figure 3) of the main flowband, with band-averages of: surface topography (thin black line) from the 2009/2011 NPI DEM, bed topography from the Fürst et al (2018) reconstruction (0-12.5 km), 2015 helicopter radar measurements (12.5-25 km), and hydrographic data in the fjord (Norwegian Mapping Authority Hydrographic 140 Service). A piecewise linear profile is fit to the bed profile for the MGM.**

## 2.2 Geometry

The bed topography of the main stream as defined in Figure 3 is not very well known. The bed topography of Svalbard glaciers has been modelled (Fürst et al., 2018) using a balance velocity approach, overridden by observed topography where radar data

are available. For Tunabreen, radar surveys were carried out by the Norwegian Polar Institute over the lower part of the glacier (Figure 3), so that the geometry of this part of the bed is reasonably well constrained; from ~16 km and downglacier, the glacier bed is below sea level, typically -40 m. The details of the upper part of the glacier bed are not dealt with; the smaller undulations in this area presumably have a limited effect on the overall dynamics of the glacier, and it is not clear how realistic all of the undulations really are. The bed undulations in the lower part of the glacier have an amplitude of typically 10 to 20 m, which is comparable to lateral (across-fjord) variations as seen in the bathymetry in front of the glacier (see map at https://toposvalbard.npolar.no; last accessed 5 December 2021). In view of the flowband geometry adapted here, the undulations are thus seen as irregularities with only a local effect on the calving process. We also note that due to depositional and erosional processes the glacier bed may be subject to significant changes even within a time span of a hundred years. These considerations led to the choice of a simple piecewise linear bed profile (red line in Figure 4).

Therefore we have:

$$b(x) = b_h - s_1 x \quad (0 \leq x \leq L_1) , \tag{8a}$$

$$b(x) = b_d \quad (x > L_1) . \tag{8b}$$

The bed profile thus drops off linearly, at rate $s_1$, until $x = L_1$, from where the bed height has a constant value of $b_d$. The parameters used here are $b_h = 650$ m; $s_1 = 0.04$; $b_d = -40$ m. It follows that $L_1 = 17250$ m.

The mean bed elevation $\bar{b}$ and mean bed slope $\bar{s}$ for given $L$ are now easily found:

$$\bar{b} = \{(b_h - s_1 L_1/2)L_1 + b_d (L - L_1)\} / L \quad (L > L_1) , \tag{9a}$$

$$\bar{b} = (b_h - s_1 L/2) \quad (L \leq L_1) . \tag{9b}$$

$$\bar{s} = (b_h - b_d / L \quad (L > L_1) , \tag{10a}$$

$$\bar{s} = s_1 \quad (L \leq L_1) . \tag{10b}$$

With these expressions the mean ice thickness as well as the surface mass budget of the main stream can be calculated for any value of $L$.

In the present model, the tributary glaciers are just considered to be buckets with a fixed geometry. When they have a positive budget they spill over and supply mass to the main glacier. The model has no intention to describe the physical process of how the tributaries flow into the main stream. Although in reality a tributary glacier with a negative mass budget can for a short period still deliver some mass, this will always be a small amount during a limited period of time. The mass budget of an individual basin is then given by

$$M_i = \beta \iint_{A_i} (h - E) dA = \beta A_i (\bar{h}_i - E) , \tag{11}$$

Where $A_i$ is the area and $\bar{h}_i$ is the mean surface elevation of basin $i$ (determined from a digital elevation model). When $M_i > 0$, the mass is added to the budget of the main stream. When $M_i \leq 0$, there is no coupling between basin and main stream. For the present geometry of the basins this happens when $E > 853, 747$ and $663$ m a.s.l. for basins 1, 2 and 3, respectively.

## 2.3 Calving rate

As in earlier studies with MGMs (Hansbreen, Oerlemans et al. 2011; Monacobreen, Oerlemans, 2018), the calving rate is assumed to be proportional to the water depth $d$. The calving flux can then be formulated as

$$F = -c \, d \, W H_f , \tag{12}$$

Here $H_f$ is the ice thickness at the glacier front, and $c$ is the 'calving parameter'. The dependence of the calving flux on water depth has been suggested, by among others, Brown et al. (1982), Funk and Röthlisberger, 1989; Pelto and Warren (1991), Björnsson et al. (2000). Recent attempts to model calving in detail have provided useful insights into the mechanical processes involved (e,g, Krug et al., 2014), but cannot simply be adopted in a more general calving law for use in large-scale models of tidewater glaciers. Recent field and remote sensing studies show that an important control on calving at many Svalbard tidewater glaciers is undercutting of the submerged ice front by melting (Petlicki et al., 2015; Luckman et al., 2015; How et al., 2019). This implies that the calving parameter depends on water temperature. However, since observations of water temperatures going further back in time are not available, we use a constant value of $c$. By matching with the mean observed calving rate for the period 2012-2019 (namely 270 m a$^{-1}$) and a typical water depth of 40 m, we find $c = 6.75$ a$^{-1}$. This may overestimate calving rates further back in time, because water temperatures were presumably lower in the 20th century. Unfortunately, it is not possible to give a quantative evaluation of the uncertainty in $c$ for longer periods, because data are not available. Also, we could not find in the literature a study in which a systematic relation between calving rate and surging phase was found.

The thickness at the glacier front is not explicitly calculated and therefore has to be parameterized. As for the studies of Hansbreen and Monacobreen referred to above, the following parameterization is used:

$$H_f = \max\{\kappa \overline{H}; \delta\} .$$  (13)

Here $\delta$ is the ratio of water density to ice density, and $\kappa$ is a constant giving the ratio of the frontal ice thickness to the mean ice thickness. For the current geometry of Tunabreen, $\kappa \cong 0.3$. So according to Eq. (13) the ice thickness can never be less than the critical thickness for flotation. For the simulations discussed in this paper, flotation was never approached. The use of Eqs. (12) and (13) allows a smooth transition between a land-terminating and tidewater glacier, a prerequisite for long-term simulations in which a model glacier should have the possibility to grow from zero volume to a long calving glacier, and backwards.

## 2.4 Imposing surges

In the MGM, surges are not internally generated but have to be prescribed. For a discussion on surging mechanisms the reader is referred to, among others, Sund et al. (2009), Mansell et al. (2012), and Benn et al. (2019). Including surges is potentially important, because they affect the mass budget and thus may exert an influence on the long-term evolution of a glacier. When a glacier surges and its front advances, the mean surface elevation decreases and the ablation zone expands. In the MGM a surge is imposed through the function $S(t)$ in Eq. (3). A fast decrease in $S$ implies a smaller ice thickness, and to fulfil mass conservation the glacier length has to increase. So the surge is in fact modelled as a sudden reduction in the mean basal resistance, without specifying its cause, where it is initiated, or how it propagates.

In the model the surge function is prescribed as (Oerlemans, 2011):

$$S(t) = 1 - S_0(t - t_0)e^{-(t-t_0)/t_s} .$$  (14)

The surge starts at $t = t_0$; two additional parameters, $S_0$ and $t_s$, determine the amplitude and the characteristic time scale of the surge. In the earlier studies of Abrahamsenbreen and Monacobreen, periodic surging was imposed, with fixed values of $S_0$ and $t_s$ according to the observed single surge of these glaciers. However, in the case of Tunabreen four surges of different amplitude and duration have been observed, which are included in the model by using different surge parameters. This allows a closer match between observed and simulated glacier length.

## 2.5 Climate forcing

Information on the climate history of Svalbard is mainly based on geomorphological and geological studies (e.g. CAPE, 2006; Bradley, 2016; Axford et al., 2017; Farnsworth et al., 2020). The general picture emerging from these studies is that of a warmer mid-Holocene climate with much reduced glacier extent over Svalbard, with gradual cooling afterwards. This cooling of 1 to 3 degrees is normally interpreted as a direct insolation effect (changing orbital parameters reducing summer radiation). Most glaciers on Svalbard reached maximum stands during the local Little Ice Age, between 1850 and 1900 (e.g. Martin-Moreno et al., 2017). Significant warming started around 1900 (Divine et al., 2011) and continues until today, but with some interruptions, notably between 1950 and 1980. On Svalbard the relation between the ELA and meteorological parameters is perhaps more complicated than for mid-latitude conditions, partly because refreezing plays a larger role. We have considered results from a detailed energy-balance model of Svalbard mass balance (Van Pelt et al., 2019) to derive model ELA values for Tunabreen, for the period 1957-2020. However, to have a useful ELA history to simulate observed glacier length, this reconstruction has to be extended backwards to at least 1900. We tried to do this by means of a reduced major axis regression on the modelled ELA to meteorological parameters observed at Longyearbyen (Nordli et al., 2020; Førland et al., 2011, with updates). The correlation with summer temperature appears to be significant (coefficient 0.46), but in the end explains only 25% of the ELA variability. In spite of this, we reconstructed the ELA history back to 1900 (referred to later as $ELA_{LYR}$). Some test calculations with the resulting forcing function were not satisfactory (this will be shown and discussed later), and therefore we decided to use for the reference experiment a forcing function of a smooth and simple form. The forcing is written as

$$E(t) = E_0 + c_1(t - 1900)^2 - c_2 e^{-((t-1975)/30)^2}.\tag{15}$$

According to Eq. (15), the equilibrium line rises quadratically in time. On this rise a minor depression of Gaussian shape is superposed, centred around the year 1975 and having a characteristic window of 30 years. In the simulation of the observed length record, the parameters $E_0, c_1$ and $c_2$ are determined in such a way that the best possible match between observed and simulated glacier length is obtained. So in effect, the climate forcing is reconstructed by inverse modelling on the glacier length observations.

## 3 Basic experiments on the sensitivity of Tunabreen to climate change

Before discussing simulations with the time-dependent climatic forcing described in section 2.5, it is useful to obtain a feeling for the sensitivity of glacier length to environmental parameters like the ELA and the calving rate. We can obtain insight into climate sensitivity and response times by doing this in numerical experiments with stepwise forcing imposed on a steady state. First of all the value of $\alpha$ in eq. (3) was obtained by matching the simulated and observed mean ice thickness (for the profile shown in Figure 4), yielding $\alpha = 1.96$ m$^{1/2}$. With $E = 524$ m a.s.l. and $c = 6.75$ a$^{-1}$, the model now produces a steady-state glacier with a length that corresponds to the situation around 1920 ($L \approx 25.8$ km). In this case the net balance of the main stream is $-0.33$ m w.e. a$^{-1}$, of the tributaries $+1.10$ m w.e. a$^{-1}$, and from calving $-0.76$ m w.e. a$^{-1}$, the latter obtained by dividing calving flux by glacier area for a good comparison. So although the main stream also has an accumulation area (Figure 3), its mean surface balance is negative and the influx from the tributaries is essential to maintain the long glacier tongue of Tunabreen.

In Figure 5 glacier length is shown for different perturbations of the ELA. All integrations start at $t = 0$, and the perturbation in the forcing is applied at $t = 1500$ a (to make sure that the initial state represents a steady state). For a 25 m rise of the equilibrium line the glacier would become about 5 km shorter, but it takes a long time to approach a steady state ($\sim 400$ years). The large climate sensitivity (defined as $\partial L/\partial E$) and long timescale is a consequence of the very small mean bed (and surface)

slope (Oerlemans, 2001, 2011). For larger perturbations of the ELA (+50 m, +100 m) the sensitivity becomes less and the response time shorter (typically ~100 a for the case with $\Delta E = +100$ m). This is related to the position of the glacier snout. When the glacier snout is still calving but on the upward sloping part of the bed (Figure 4), the dependence of the calving flux on the water depth makes the glacier more stable (a smaller change in $L$ is needed to achieve the necessary change in the mass budget to acquire a new equilibrium state). The height – mass balance feedback becomes particularly effective for negative values of $\Delta E$. Because the calving rate is constant, due to constant water depth, there is no way for the model glacier to stabilize. In reality, the advance would come to a halt because the glacier front would eventually encounter deeper water and higher water temperatures.

The red curves in Figure 5 shows the response of the glacier to changes in the calving parameter $c$. For a 50 % larger calving parameter, the response of the glacier is comparable to that for a 35 m increase in the ELA. For a 25% decrease in the calving rate, the glacier grows slowly, but steadily.

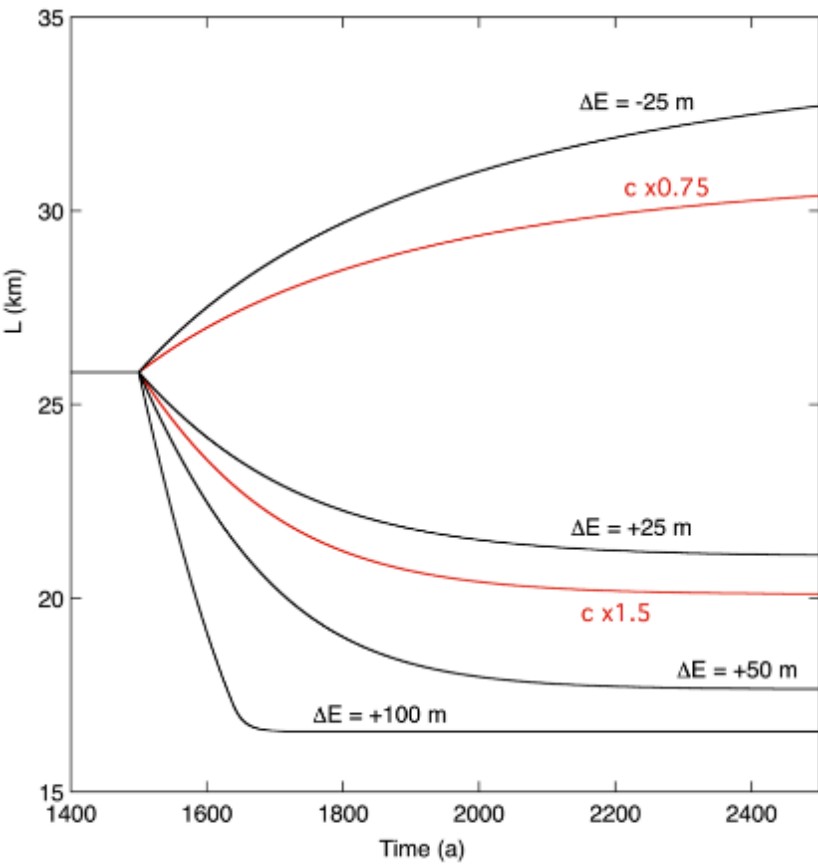

**Figure 5. Evolution of glacier length for different perturbations of the ELA ($\Delta E$), and different values of the calving parameter $c$.**

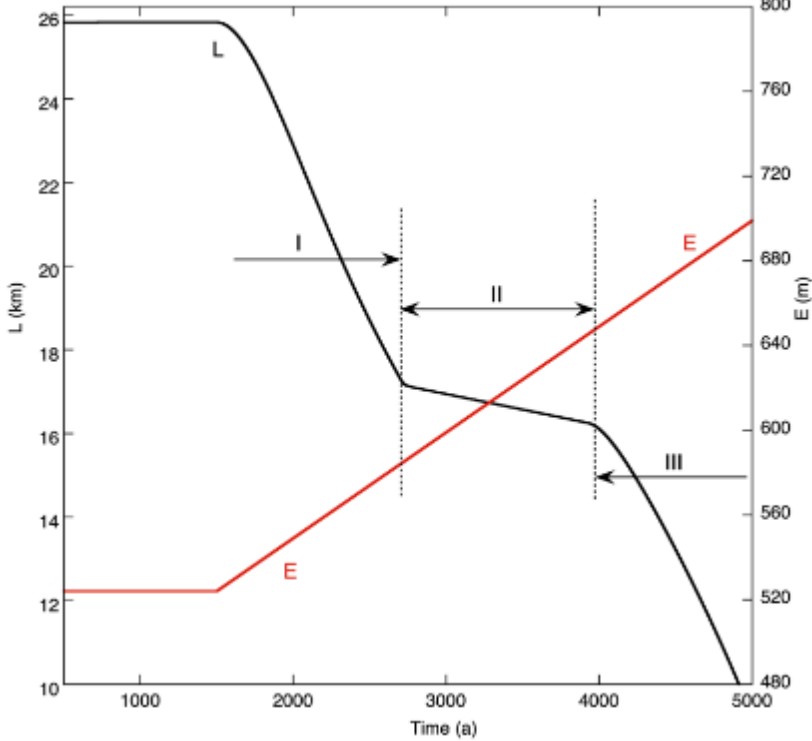

**Figure 6. Glacier length $L$ for a very slowly increasing ELA ($E$). Three regimes are identified (I, II, III) for which the sensitivity of glacier length to the ELA ($\partial L/\partial E$) differs significantly.**

The apparent differences in the sensitivity for smaller and larger perturbations of the ELA call for a further numerical experiment in which this is explored. One way to approach this is to do a long integration with very 'slow' forcing (here slow means that the glacier is always close to an equilibrium state). Figure 6 shows the result of an integration in which the ELA increases at a rate of $0.05$ m a$^{-1}$. Evidently, with respect to the climate sensitivity $\partial L/\partial E$ (a dimensionless quantity), three regimes can be distinguished. In regime I the glacier front is calving and located on the part of the bed with constant water depth. The mean climate sensitivity in this regime is $\partial L/\partial E \approx 150$. As noted before, this very large value is a direct consequence of the small mean bed slope (strong height - mass balance feedback). In regime II the glacier front is on the part of the bed where the water depth decreases when going inland. When the glacier becomes shorter to adapt to the increasing ELA, the calving rate decreases strongly (according to Eq. (12)). A very small change in $L$ is therefore sufficient to restore equilibrium, and consequently the climate sensitivity is small ($\partial L/\partial E \approx 15$). In regime III the glacier has become a land-terminating glacier. The combined effect of a larger mean bed slope and the absence of calving results in a climate sensitivity comparable to that for regime I (now $\partial L/\partial E \approx 140$). This calculation thus illustrates once more how important geometric factors are when considering the response of individual glaciers to climate change.

## 4 Simulating the evolution of Tunabreen during the past 100 years

We now turn to simulation of the observed length record. The best possible result for optimal values of the surge parameters ($t_0, S_0, t_s$) and climatic parameters ($E_0, c_1, c_2$) is shown in Figure 7. The value of $E_0$ was set to a value that makes the simulated glacier length in 1924 equal to the observed value ($E_0 = 490$ m a.s.l.). The surge amplitudes have been chosen such that the advance during the surge approximately matches the observed length change.

During the 1924 surge, which has the largest amplitude ($\Delta L \approx 3$ km), the associated maximum reduction in the mean ice thickness (and surface elevation) is about 25 m. Although the calving rate is constant, the calving flux decreases slightly because the frontal ice thickness is somewhat smaller. Altogether, at the end of the surge the net mass budget perturbation of

295 the main stream is about -0.24 m w.e. a$^{-1}$ and of the entire glacier about $-0.18$ m w.e. a$^{-1}$. It should be noted that there are two effects leading to a negative balance perturbation: (i) the lowering of the glacier surface, and (ii) the extension of the ablation zone. Together with the climatic forcing this affects the retreat of the glacier front after the surge.

For the 1966-1971 surge the simulated retreat is somewhat too fast. However, increasing the surge parameter $t_s$ does not help because it leads to a glacier length that is too large when the next surge starts. The last surge comes fast and has a small

amplitude, but it was possible to choose the surge parameters in such a way that the 2020 observed length is reproduced. The dotted line in Figure 7 shows the result for a run without surging. In this case the glacier length stays close to the minimum values after each surge except for the last surge. The 2016 surge comes so fast after the previous surge that the glacier has no time to adjust its length to the higher ELA. This is reflected in the difference between the change in length over the past 100 years ($\approx$2 %) as compared to the change in volume ($\approx$11 %). The decrease in mean glacier length has thus been limited due to

the increasing surge frequency, at the cost of a reduction of the mean ice thickness.

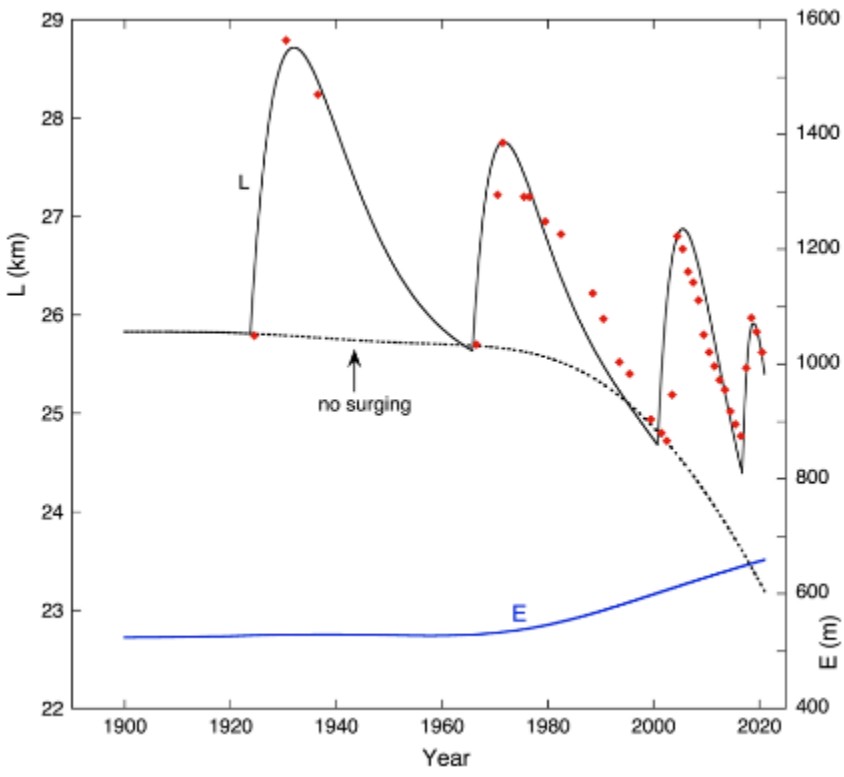

**Figure 7. Simulation of glacier length *L* achieved after optimization of the model parameters. Observations are indicated by red dots. The blue line shows the optimal ELA forcing (scale at right). Glacier length simulated for a case without surging is shown by**

310 **the dotted line.**

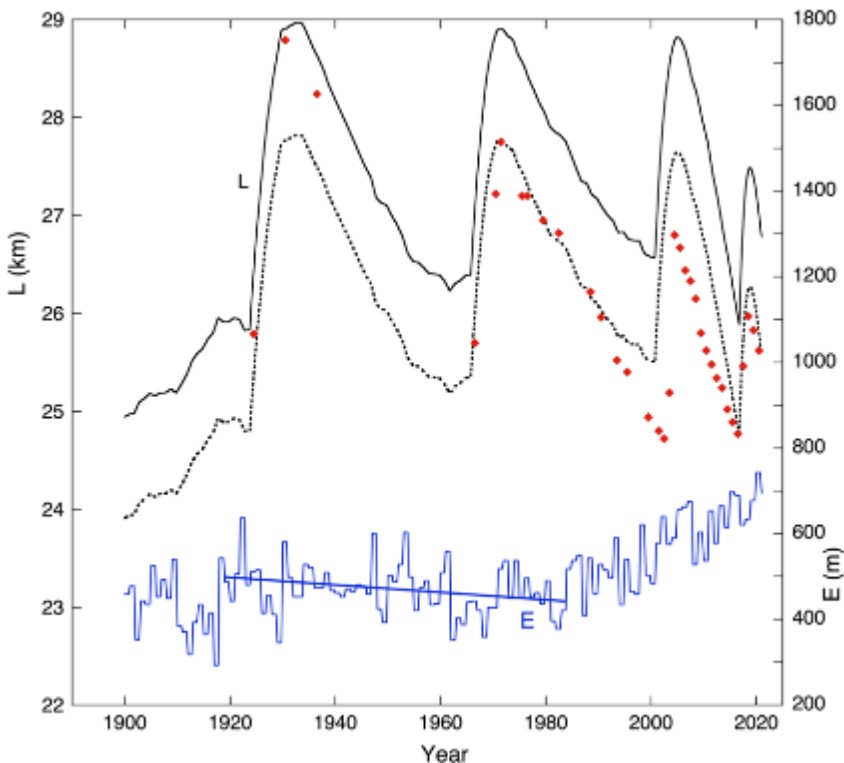

**Figure 8. Simulation of glacier length $L$ with $ELA_{LYR}$ as climate forcing (scale at right). Observations are indicated by red dots. The simulation shown by the solid line is tuned to the first observed glacier length (1926), the dotted line is tuned to the last observed glacier length (2020).**

The result of Figure 7 has been obtained with the following parameters of the ELA history: $E_0 = 524$ m a.s.l., $c_1 = 0.0095$ m a$^{-2}$, $c_2 = 40$ m. Changing these parameters leads to a larger difference between observed and simulated glacier length. This cannot be compensated by adjusting the surge parameters, i.e. the parameter sets $(t_0, S_0, t_s)$ and $(E_0, c_1, c_2)$ play fairly independent roles in the model. The ELA history that delivers the best simulation is therefore well determined and reveals that a 135 m increase in the ELA over the past 50 years is sufficient to explain the behaviour of Tunabreen.

Two Digital Elevation Models (DEMs) for Svalbard have recently been published, referring to the years 1936 and 2010 (Geyman et al., 2022). The DEMs allow an independent check on the model performance. The mean change in surface elevation for Tunabreen over the period 1936 – 2010, based on gridded DEM data, amounts to $-0.30$ m w.e. a$^{-1}$. The mean net balance calculated for this period from the model output (run of Fig. 7) is $-0.25$ m w.e. a$^{-1}$. We thus conclude that the calibrated model result is in broad agreement with the geodetic evidence.

At this point it is interesting to return to the ELA reconstruction based on energy-balance modelling and the Longyearbyen meteorological record as discussed in section 2.5 ($ELA_{LYR}$). In Figure 8 computed glacier length is shown for the $ELA_{LYR}$ forcing. Two simulations are shown: one in which the first data point (1926) is matched with the observed length, one in which the last data point (2020) is matched. The corresponding value of $E_0$ are 528 and 533 m. None of the simulations are good, and adjusting the surge parameters does not give an improvement. The reason for the discrepancy between observed and simulated glacier length is the small but significant decline of the ELA during the period 1920 – 1980. When the equilibrium line starts to rise around 1985, the response of the glacier is too slow to catch up with the observed retreat over the last hundred years. It thus appears that the value of $ELA_{LYR}$ as a climate proxy for Tunabreen is limited.

The mass-balance simulation with a regional climate model by Van Pelt et al. (2019) yields a mean increase of 4.6 m a$^{-1}$ of the ELA over the period 1957-2018. This is substantially larger than the 3.2 m a$^{-1}$ found for the reconstructed forcing. The

discrepancy is substantial and hard to explain. However, since the simulation by Van Pelt et al. (2019) does not go further back in time than 1957, a thorough comparison remains difficult.

Referring to the sensitivity of the glacier length to the calving parameter (Figure 5), it is obvious that the use of a fixed calving parameter is a limitation of the calibration procedure. It would even be possible to keep the ELA fixed and vary $c$. Possibly variations in $c$ and the *ELA* work in parallel, because both quantities presumably increase with atmospheric temperature (assuming that water temperature in summer is related to air temperature). However, increasing $c$ would imply a smaller increase in the ELA, and the discrepancey between the reconstructed ELA history and the simulation by Van Pelt et al. (2019) would become larger.

## 5 The future evolution of Tunabreen

Having calibrated the model with observations over the past hundred years, we now consider future climate change scenarios and see how Tunabreen might change in the coming hundred years. Rather than applying the output from regional climate model simulations with all their uncertainties, we prefer a simple approach in which the equilibrium line rises at a constant rate after the year 2020.

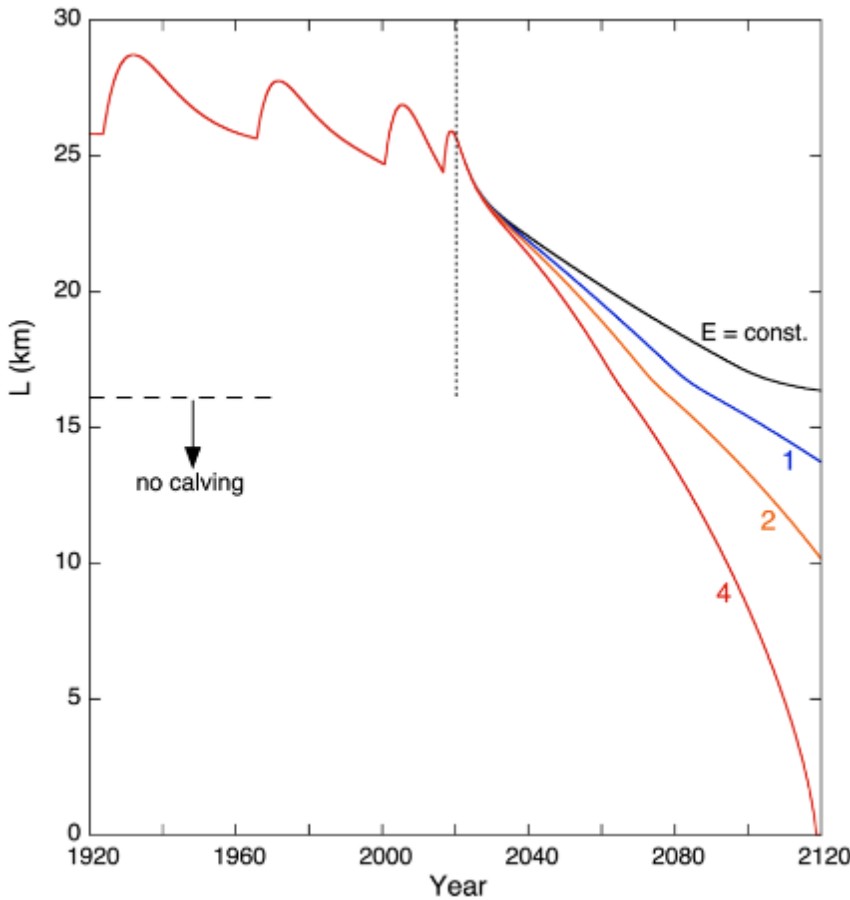

**Figure 9. Glacier length $L$ for different climate change experiments. The black curve refers to a calculation in which $E$ is kept constant at its 2000 - 2019 mean value. The colour labels refer to the imposed constant rate of change of $E$ per year from 2020 onwards.**

In Figure 9 some results of climate change experiments for the next hundred years are summarized. In the reference run the ELA is kept constant at the 2020 value from the reconstruction, namely 656 m a.s.l. In this case the model glacier retreats

during the next hundred years at an almost linear rate of ~80 m a$^{-1}$, which illustrates by how much the current glacier stand is out of equilibrium with the present climate. After the year 2110 the retreat slows down and the glacier approaches a new steady state with a length of about 16 km and no calving anymore.

For the intermediate climate warming scenario (the equilibrium line then rises by 2 m a$^{-1}$), Tunabreen would retreat by about 15 km over the next hundred years. During the retreat the net balance would steadily decrease to about $-2.0$ m w.e. a$^{-1}$, i.e. the glacier is getting more and more out of balance. In the year 2026 basin 3 would no longer deliver mass to the main stream, for basin 2 this happens in the year 2063.

For the case of a 4 m a$^{-1}$ rise in the equilibrium line, the retreat is obviously much stronger and by the year 2120 Tunabreen
has almost disappeared. In the year 2081, $E = 900$ m a.s.l. and virtually the entire glacier is below the equilibrium line.

It can be seen in all the simulations that the retreat slows down when the length becomes less than 17.2 km, i.e. at the point where the modelled bed elevation starts to increase from -40 m to sea level (at x=16.2 km), see Figure 9. This is in line with the result of Figure 6.

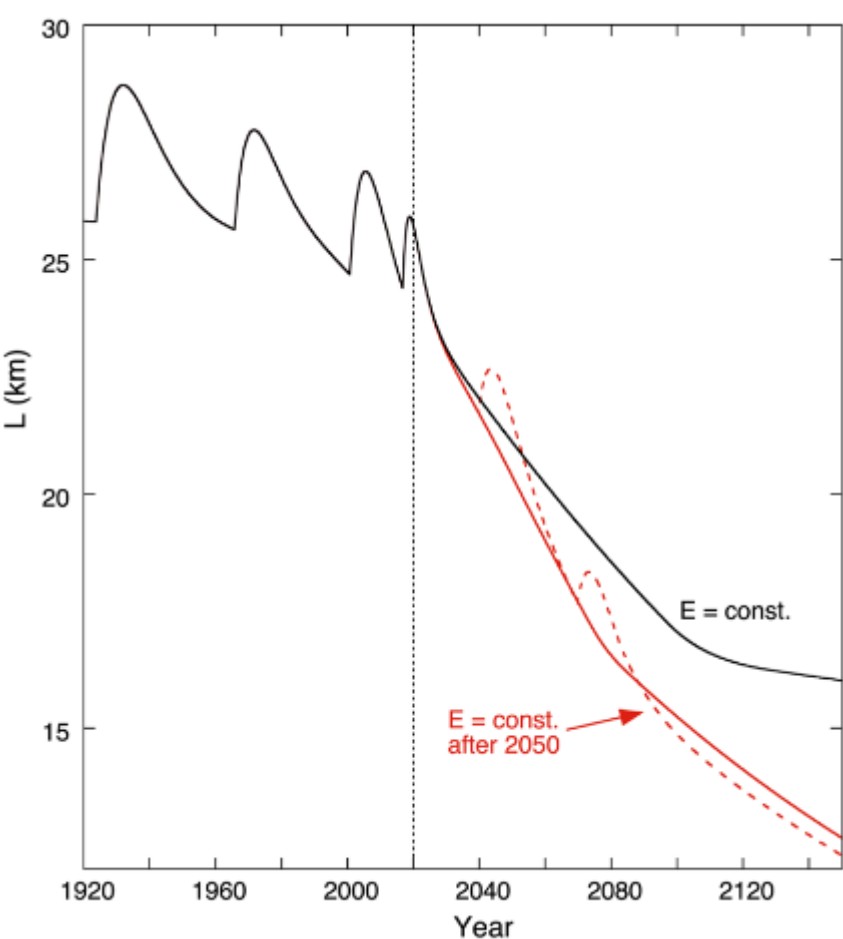

**Figure 10. Glacier length *L* for the 'Paris run' (constant ELA after 2050), shown in red. The black curve refers to the calculation in which *E* is kept constant at its 2020 reconstructed value (as in Figure 9). The dashed curve represents the 'Paris run' with two surges imposed, starting in 2040 and 2070.**

In Figure 10 a calculation is shown for an optimistic climate change scenario, in which the equilibrium line rises by 2 m a$^{-1}$ until the year 2050 and remains constant afterwards (we refer to this calculation as the 'Paris run'). We show the result until the year 2150, to see if by this time the glacier would have reached a new steady state. Obviously, this is not the case, although

the rate of retreat slows down. This case clearly illustrates that it takes a long time before limiting climate warming has a

380 significant effect on the retreat of large glaciers.

The dashed curve in Figure 10 represent the result for the 'Paris run' with two surges imposed, initiated at the years 2040 and 2070. The choice of those years is quite arbitrary of course, but it serves to demonstrate that surges do not seem to have a impact on the long-term evolution of the glacier.

To further illustrate the interaction between climate change and the mass budget of Tunabreen, we show in Figure 11 the

385 various mass balance components as a function of time for the 'Paris run' (without additional surges). All terms in the mass budget equation have been converted to specific balance rate for an easy comparison, e.g. the calving flux has been divided by the glacier area. First of all, as noted before, the balance of the main stream (*bstream*) is always negative and the glacier lives on the contributions from the tributaries (*btrib*). After 2048 the specific contribution from the tributaries becomes zero and the model glacier basically is a body of ice melting away. After 2073 the calving starts to decrease and by 2084 it has become

zero.

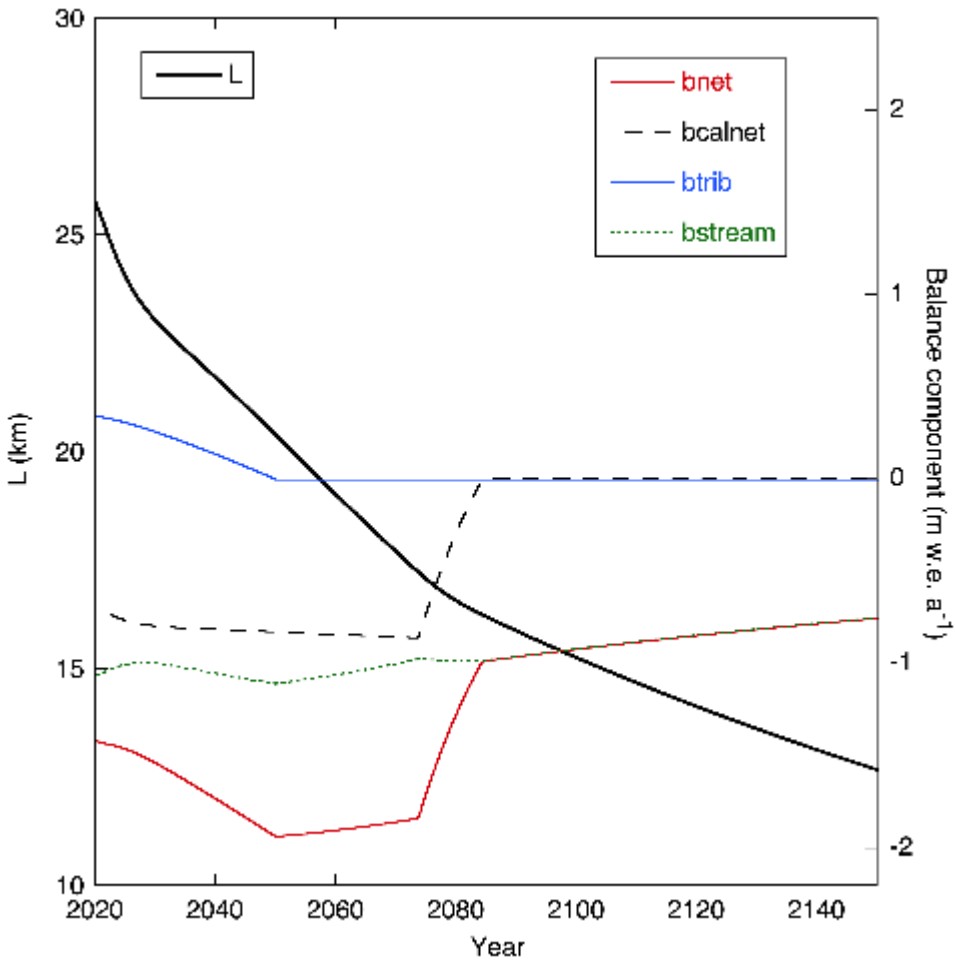

**Figure 11. Glacier length (solid black line, scale at left) and mass balance components (scale at right) for the 'Paris run'.**

## 5 Discussion

Tunabreen is a special glacier because of its complex surging behaviour, which poses a real challenge for a modelling study. The MGM, based on the view that the evolution of a glacier is primarily determined by the exchange of mass with its surroundings, offers a relatively simple tool to study the response of Tunabreen to changing environmental conditions. The application is straightforward and requires as geometric input only a topographic map and a schematic bed profile along the main flowband. We are aware of the limitations inherent to a simple modelling approach as employed here. The dynamic interaction between the tributaries and the main stream is not explicitly described and the calving process is formulated in a simple way. Since there is no spatial resolution, subtle effects associated with undulations in the bed cannot occur. Therefore, in the end one would like to repeat the simulations presented here with a comprehensive glacier flow model with full spatial resolution. However, it will not be an easy task to prepare the necessary input fields, formulate boundary conditions in a straightforward way, and get the calving and surging to occur at the right place and time!

As has been found for other larger glaciers in Svalbard, Tunabreen appears to be very sensitive to climate change. The common factor is the small surface slope, also when considered in a global perspective. From a purely geometric argument, $\partial L/\partial E \sim -2/\bar{s}$ (Oerlemans, 2001). The mean slope of Tunabreen is 0.031, comparable to that of many other glaciers in Svalbard like e.g. Monacobreen (0.025), Kongsvegen (0.032), Kronebreen (0.025). A characteristic value of $\partial L/\partial E$ therefore is -65, i.e. a 100 m rise of the equilibrium line implies a glacier retreat over 6.5 km (when a steady state would be reached). For larger mid-latitude mountain glaciers values of $\partial L/\partial E$ are in the range of -10 to -30, while for long glaciers in Alaska and Patagonia typically in the range of -30 to -60. Although other factors than mean slope play a role, it is fair to conclude that the larger glaciers on Svalbard are the most sensitive in the world. This is in line with the idea that during the mid-Holocene Climatic Optimum the degree of glacierization in Svalbard was considerably smaller than today (e.g. Fjeldskaar et al., 2017).

Our numerical experiments suggest that typical fluctuations in calving rates and equilibrium line altitudes have comparable effects on the evolution of Tunabreen. A 50 % increase in the calving rate has the same effect as a 35 m increase of the ELA (Figure 6). It is conceivable that in a warming climate with increasing ocean temperatures, calving rates will become larger. This implies that the glacier retreat curves shown in Figure 9 represent conservative estimates. However, further studies on the relation between ocean temperatures and calving rates on multi-annual time scale are needed to make a meaningful estimate of how calving might enhance the retreat of Tunabreen in the near future.

We did not find a significant impact of surging on the evolution of Tunabreen. Although a surge initiates a negative perturbation of the mass budget for some years and reduces the glacier volume slightly, this apparently has no lasting effect. The same conclusion was reached in earlier studies of Abrahamsenbreen (Oerlemans and Van Pelt, 2015) and Monacobreen (Oerlemans, 2018), and it probably applies to most long glaciers on Svalbard. However, with respect to ice caps that have surging parts (like Austfonna), the situation may be different.

**Author contribution**

JO developed the model code and performed the simulations. JK analysed and evaluated data on bedrock topography and glacier length. AL derived and compiled data on calving rates. JO prepared the manuscript with contributions from all co-authors.

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
