# Peer review of "Modelling the mass budget and future evolution of Tunabreen, central Spitsbergen"

_The Cryosphere, 2021_

## Referee Comment (RC1)

Paper No.:      tc-2021-155

Author(s):      Johannes Oerlemans, Jack Kohler, Adrian Luckman

Title:          Modelling the mass budget and future evolution of Tunabreen, central Spitsbergen

Referee:        Francisco Navarro

**GENERAL COMMENTS:**

The authors describe and discuss in the paper the future evolution of the mass budget of Tunabreen, a tidewater glacier in Svalbard that has experienced various recent surges with increasing frequency. This is done by using a so-called minimal glacier model (MGM), in which dynamic processes are parameterized. The focus of MGMs is on the exchange of mass with the environment (atmosphere and ocean). Simple models such as the MGMs have been claimed to have the advantage to allow exploring the parameter space in great detail. However, over-parameterisation of processes has also clear disadvantages, as only a minor part of the physics of the system being modelled is being represented by means of physically-based equations. Aside from this, the model presented in the paper has additional limitations (and a bias) which are either not discussed or not sufficiently discussed. My review will focus on these aspects, as well as on some shortcomings such as a certain lack of regional contextualization of the glacier under study (and its surges). The main general comments follow.

1) CONTEXT: The glacier characteristics, with emphasis on those related to surging, should be put into the context of the Svalbard glaciers:
   - Is it a large/medium/small tidewater glacier compared to the rest of Svalbard tidewater glacier?
   - Is it surface slope within the usual range?
   - Are its typical calving rates similar to those typical of other Svalbard glaciers (or higher/lower than usual)?
   - How many known surging glaciers have been identified in Svalbard?
   - What is the usual range of surging periods?
   - Do usually glacier surges in Svalbard initiate at the front and propagate upwards?
   - Has any other Svalbard glacier known to have experienced surges with increasing frequency?

2) SIMPLIFIED BED GEOMETRY: The real glacier bed geometry is approximated by a simplified geometry consisting of a flat portion below sea level and an inclined portion (with positive slope upwards) nearly all above sea-level. Several aspects should be discussed here:
   - Why the authors did not consider another flat portion in the uppermost part of the glacier? (as suggested by Fig. 4)
   - Making flat the submerged part of the bed will have an effect on the advance/retreat rates (for instance, fast retreat on reverse-bed slopes) and, most importantly, on the stability conditions, i.e., the modelled behaviour could differ significantly from the real glacier behaviour.
   - In several occasions along the text (this will be pointed out in the specific comments) references are made in the discussion of the model results to the fact that the glacier terminus in on the deeper or in the shallower part of the submarine

bed. Such comments are meaningless (as comments referred to the model results) because the simplified bed geometry has a constant water depth (ice thickness below sea level) for its submerged part.

3) BIAS IN THE MODEL: A goal of the model presented in this paper is "to determine how sensitive the glacier is to ongoing and future climate warming". The interaction with the climate is established via a ELA evolution. For (recent) past climate, a ELA based on the temperature record at Longyearbyen does not provide acceptable results ("the correlation with summer temperature … explains only 25% of the ELA variability") and therefore the authors use instead a forcing function (for the ELA as a function of time) whose parameters are calibrated against the glacier length observations ("the climate forcing is reconstructed by inverse modelling on the glacier length observations"). Later, when the (forward) model is run under this climate forcing, it is claimed that "the simulated glacier retreat is in good agreement with observations". Was something else expected, having calibrated the ELA (climate) history with the observed length fluctuations? There is a clear bias in the model, which makes the model results of limited value.

If the temperature record at Longyearbyen could not be used as a proxy for the ELA fluctuations in Tunabreen, theoretically the wisest approach (though very costly) would have been the use of a regional climate model (in hindcast mode), downscaled for Tunabreen and combined with a mass balance model to reconstruct the ELA history. I am aware, anyway, that the necessary data for the downscaling of the RCM and for the mass balance model are not available, so neither this approach is realistic. Nor the use of reanalysis data solves the problem, as such data are only available since around 1950 (and we still would have downscaling difficulties). In other words, the authors have done probably the only thing that they could do. But not because of this we have to be "permissive" with their model results but, on the contrary, be somehow skeptic and keep always in mind the model limitations implied by this bias.

4) MODEL LIMITATIONS:

The authors should discuss in the most suitable place (either the corresponding paragraph, the introduction or the discussion sections) the main limitations of the model employed. In particular:

a) Limitations inherent to MGMs: it would be good that the authors would briefly discuss the main limitations of the MGMs.

b) Simplified bed geometry (already discussed in general comment 2).

c) Bias involved in the climate forcing (already discussed in general comment 3).

d) Tributary basins with fixed geometry. Considering a fixed geometry (including surface geometry) for the tributary basins poses some problems. It is correct that, if the surface geometry of the tributaries is assumed to be constant, then the supply of mass from a given tributary to the main trunk will be given by the net surface mass balance on the tributary (as stated by eq. (11)). However, assuming a fixed surface geometry for the tributaries when the main trunk's geometry varies with time in not very realistic (it is equivalent to considering a "step" in the surface geometry when passing from the tributary to the main trunk). Moreover, $M_i \leq 0$ is not physically possible, because, as long as the tributary has some dynamics, there will always be a supply of mass by advection from the tributary to the main trunk. In the "real world" they could become uncoupled (physically disconnected) when the main trunk terminus would retreat upglacier past the point of junction

with the tributary. But this is also physically inconsistent with a constant surface geometry of the tributaries.

e) Prescribing surges prevents the model to be applied for predictions (although the conclusion that the surge occurrence does not have long-lasting effects on the glacier front position makes this not so relevant – but at least should be cited).

f) A constant calving rate does not seem a suitable choice for a frequently surging glaciers, as surges imply more intense frontal crevassing and therefore more frequent calving events.

Many of the model assumptions leading to the above limitations could be maintained in the paper, but they should be discussed as described so the readers are fully aware of the limitations inherent to the model and its results.

**SPECIFIC COMMENTS:**

ABSTRACT:

- P1, L16-18: As the ELA history is reconstructed by matching observed and simulated glacier length because there was a modest correlation between Tunabreen's ELA and Longyearbyen's temperature, I would suggest altering the order of the two sentences in lines 16-18.

- P1, L19: As this would be expected due to the bias mentioned in general comment 3, I would try to change the sentence so it does not seem a substantial finding.

- P1, L23: becomes -> would become?

1) INTRODUCTION:

A main modification is this section is to describe Tunabreen and its surges in the context of Svalbard glaciers and surges, as described in general comment 1.

- P2, Figure 2: Perhpas it would be worth noting somewhere that the front positions before the surges (for the first two surges around 25.7-25.8 km; for the last two surges, around 24.7-24.8 km) all are located in the bed bump that can be appreciated at 24-26 km in Fig. 4 (this is only relevant to observations, not to the modelling results, because of the assumption of a flat submarine bed).

- P3, L53: "effect of reverse bed slopes" and "variable calving rates" are both cited here among the feedback mechanisms that can be dealt with by using MGMs. However, these two are not considered in the present paper. The sentence is correct, as it is general for MGMs, but could induce the reader to think that all of these feedback mechanisms will be considered in the paper. Try to think in a writing that avoids this (or perhaps just say that these two particular ones will not be considered in the present paper).

2) GLACIER MODEL:

- P4, L78, eq. (1): perhaps a reader not fully familiar with the mass balance terminology and units could suspect an inconsistency in the dimensions of the various terms in the equation (time derivatives of volume together with [apparent] masses). This could be avoided by describing the units of the various variables involved. Also, in line 79 it would be good to refer to $F$ as *volumetric* calving flux (to distinguish from mass flux) and stating that it is expressed as an specific value over the glacier area (i.e., volumetric calving flux divided by the glacier area).

2.1 *Prognostic equations for glacier length*:

- P4, L89: Some text is missing here (likely, definitions of $N$, $L$, $H_m$ and "Deriving").
- P4, L93, eq. (3): This is eq. (4.2.1) in Oerlemans (2011). Although Oerlemans (2011) is easily downloadable through the Internet, it would be convenient to add here a couple of lines justifying/explaining this parameterization.
- P4, L98: Add a reference here supporting this statement.
- P5, L111, eq. (6): State that the variable $h$ is surface elevation.

2.2 *Geometry*:

Important regarding this section are the aspects discussed in general comment 2.

- P6, L144: Similarly, limitations regarding the fixed surface geometry of the tributaries discussed in general comment 4d should be mentioned here.

2.3 *Calving rate*:

- P6, L150-153: I suggest changing "the dominant control" by "a dominant control", as other controls have been shown to exert an important role (e.g. meltwater filling the crevasses – or enlarging them when flowing downwards through crevasses –, effect of ice mélange), and add to the references listed De Andrés et al. (2018, 2021), doi: 10.1017/jog.2018.61 and 10.1017/jog.2021.27, respectively.
- P6, L155-156: limitations regarding the constant calving rate discussed in general comment 4f should be mentioned here (with greater detail than that used there).
- P6, L162: some text is missing here.

2.4 *Imposing surges*:

Limitations regarding the prescription of surges discussed in general comment 4e should be mentioned here.

2.5 *Climate forcing*:

The bias discussed in general comment 3 should be discussed in this section.

- P8, L202: it was decided -> we decided
- P8, L202-203: the sentence "in line with the statement in the beginning of this section" is ambiguous (I imagine that it refers to that on lines 192-193, but the reader should understand it clearly).

3) BASIC EXPERIMENTS ON THE SENSITIVITY OF TUNABREEN TO CLIMATE CHANGE:

- P8, L226 and P9, Figure 5: It would be convenient to comment the case $\Delta E=+100$ m in relation with Fig. 5 and the fact that, in this case, instead of a smooth curve (as in other cases shown in the figure) there is a sharp change some earlier than 1700 yr.
- P8, L232: shows -> show
- P9, Figure 6: It would be better to restrict Regime I to the section starting in year 1500 (and ending close to year 3000); Before year 1500 it dos not make much sense.
- P10, L249-257: The comments in regimes I and II regarding water depth are meaningless under the assumption that the model is using a simplified geometry with flat bed in its submerged part (see general comment 2).

4) SIMULATING THE EVOLUTION OF TUNABREEN DURING THE PAST 100 YEARS:

- P10, L260: Some text is missing here.

- P10, L263-275: The 2002-2004 surge is the only one not discussed.
- P10, L273: I guess that "large part" should be "last one". Additionally, is there an explanation for the statement in the sentence?

5) THE FUTURE EVOLUTION OF TUNABREEN:

- P13, L313-314: "a glacier lenght of 16.2 km and no calving anymore" is stated here, but in line 136 $L_1$=17250 m is mentioned.
- P13, L316: I suspect that the last $m^{-1}$ (at the end of the line) is a typo.
- P13, Figure 10: Two surges have been imposed, starting in 2030 and 2065. Explain how these values were selected in the light of the dating/frequencies of the previous history of (observed) surges.
- P14, L341: Again, the mention that "the front comes into shallow water" is meaningless in the context of a model with simplified geometry with constant thickness of submerged part (i.e., "shallow water" – versus deeper water – does not exist in the model).

6) DISCUSSION:

The main point here would be the need to summarize the main limitations of the model (as discussed earlier) and how the model results should be taken with caution.

- P15, L359: Add reference to Oerlemans at the end of the first sentence (although mentioned earlier – line 355 –, it will make it clearer).

---

## Author Comment (AC1)

**NOTE: Quotations from review in normal face. Our response in *italic***

**GENERAL COMMENTS:**

The authors describe and discuss in the paper the future evolution of the mass budget of Tunabreen, a tidewater glacier in Svalbard that has experienced various recent surges with increasing frequency. This is done by using a so-called minimal glacier model (MGM), in which dynamic processes are parameterized. The focus of MGMs is on the exchange of mass with the environment (atmosphere and ocean). Simple models such as the MGMs have been claimed to have the advantage to allow exploring the parameter space in great detail. However, over-parameterisation of processes has also clear disadvantages, as only a minor part of the physics of the system being modelled is being represented by means of physically-based equations. Aside from this, the model presented in the paper has additional limitations (and a bias) which are either not discussed or not sufficiently discussed. My review will focus on these aspects, as well as on some shortcomings such as a certain lack of regional contextualization of the glacier under study (and its surges). The main general comments follow.

> *We regret that the reviewer does not have more appreciation of the approach we take. It is of course true that more comprehensive models with spatial resolution can deal with more physical processes in detail. However, we believe that higher-order models are overvalued, and that many problems involved in formulating the boundary conditions or the various tricks required to keep the numerical scheme stable are often swept under the carpet. It has NEVER been demonstrated that higher-order models do a better job in simulating glacier records that are 100 to 150 years long, compared to simple models. In our view, when the interest is in long-term evolution, it is more natural to consider a glacier as a damped mass-budget system with inputs and outputs, rather than as a pile of subtle ice mechanical processes.*

1) CONTEXT: The glacier characteristics, with emphasis on those related to surging, should be put into the context of the Svalbard glaciers:
• Is it a large/medium/small tidewater glacier compared to the rest of Svalbard tidewater glacier?
• Is it surface slope within the usual range?
• Are its typical calving rates similar to those typical of other Svalbard glaciers (or higher/lower than usual)?
• How many known surging glaciers have been identified in Svalbard?
• What is the usual range of surging periods?
• Do usually glacier surges in Svalbard initiate at the front and propagate upwards?
• Has any other Svalbard glacier known to have experienced surges with increasing frequency?

> *Our paper is not intended to be a general paper about surging, but is instead focused on providing a model of a glacier that has undergone some surges. Here we face the usual conflict between a journal that wants a paper to be compact and concise, and a reviewer that wants to see more extensive descriptions.*

> *However, we will add a few lines to provide more context.*

2) SIMPLIFIED BED GEOMETRY: The real glacier bed geometry is approximated by a simplified geometry consisting of a flat portion below sea level and an inclined portion (with positive slope upwards) nearly all above sea-level. Several aspects should be discussed here:
– Why the authors did not consider another flat portion in the uppermost part of the glacier? (as suggested by Fig. 4)

– Making flat the submerged part of the bed will have an effect on the advance/retreat rates (for instance, fast retreat on reverse-bed slopes) and, most importantly, on the stability conditions, i.e., the modelled behaviour could differ significantly from the real glacier behaviour.

– In several occasions along the text (this will be pointed out in the specific comments) references are made in the discussion of the model results to the fact that the glacier terminus in on the deeper or in the shallower part of the submarine bed. Such comments are meaningless (as comments referred to the model results) because the simplified bed geometry has a constant water depth (ice thickness below sea level) for its submerged part.

*The bed undulations in the lower part of the glacier have an amplitude of typically 10 to 20 m, which is comparable to lateral (across-fjord) variations as seen in the bathymetry in front of the glacier (see map at https://toposvalbard.npolar.no; last accessed 2 September 2021). In view of the flowband geometry adapted here, the undulations are thus seen as irregularities with only a local effect on the calving process. We also note that due to depositional and erosional processes the glacier bed may be subject to significant changes even within a time span of a hundred years.*

3) BIAS IN THE MODEL: A goal of the model presented in this paper is "to determine how sensitive the glacier is to ongoing and future climate warming". The interaction with the climate is established via a ELA evolution. For (recent) past climate, a ELA based on the temperature record at Longyearbyen does not provide acceptable results ("the correlation with summer temperature ... explains only 25% of the ELA variability") and therefore the authors use instead a forcing function (for the ELA as a function of time) whose parameters are calibrated against the glacier length observations ("the climate forcing is reconstructed by inverse modelling on the glacier length observations"). Later, when the (forward) model is run under this climate forcing, it is claimed that "the simulated glacier retreat is in good agreement with observations". Was something else expected, having calibrated the ELA (climate) history with the observed length fluctuations? There is a clear bias in the model, which makes the model results of limited value.

If the temperature record at Longyearbyen could not be used as a proxy for the ELA fluctuations in Tunabreen, theoretically the wisest approach (though very costly) would have been the use of a regional climate model (in hindcast mode), downscaled for Tunabreen and combined with a mass balance model to reconstruct the ELA history. I am aware, anyway, that the necessary data for the downscaling of the RCM and for the mass balance model are not available, so neither this approach is realistic. Nor the use of reanalysis data solves the problem, as such data are only available since around 1950 (and we still would have downscaling difficulties). In other words, the authors have done probably the only thing that they could do. But not because of this we have to be "permissive" with their model results but, on the contrary, be somehow skeptic and keep always in mind the model limitations implied by this bias.

*Why is it so clear to the reviewer that there is a bias in the model, rather than in the representativeness of the observations, or in the meteorological model used to simulate glacier mass balance? The simulation of glacier mass balance with climate models over LONGER periods of time (>100 years) has hardly been done, because it is problematical. In a recent study concerning an alpine glacier (Tschierva Glacier, Switzerland; Oerlemans et al, 2021, Journal of Glaciology) it was shown that it is just impossible to simulate an observed glacier length record by forcing a glacier model with climate model output.*

*We agree that the statement "the simulated glacier retreat is in good agreement with observations" is a bit silly. However, reconstructing the forcing from the length record is the only way to arrive at a proper initial state for integrations into the future. In our view, the sentence "which makes the model results of limited value" is not justified.*

4) MODEL LIMITATIONS:

The authors should discuss in the most suitable place (either the corresponding paragraph, the introduction or the discussion sections) the main limitations of the model employed. In particular:

a) Limitations inherent to MGMs: it would be good that the authors would briefly discuss the main limitations of the MGMs.

*We feel the description is quite adequate and states clearly the philosophy of MGMs. But we will add a few more lines.*

b) Simplified bed geometry (already discussed in general comment 2).

*See text above about undulations.*

c) Bias involved in the climate forcing (already discussed in general comment 3).

*See above*

d) Tributary basins with fixed geometry. Considering a fixed geometry (including surface geometry) for the tributary basins poses some problems. It is correct that, if the surface geometry of the tributaries is assumed to be constant, then the supply of mass from a given tributary to the main trunk will be given by the net surface mass balance on the tributary (as stated by eq. (11)). However, assuming a fixed surface geometry for the tributaries when the main trunk's geometry varies   with time in not very realistic (it is equivalent to considering a "step" in the surface geometry when passing from the tributary to the main trunk). Moreover, $MM \leq 0$ is not physically possible, because, as long as the tributary has some dynamics, there will always be a supply of mass by advection from the tributary to the main trunk. In the "real world" they could become uncoupled (physically disconnected) when the main trunk terminus would retreat upglacier past the point of junction with the tributary. But this is also physically inconsistent with a constant surface geometry of the tributaries.

*The tributary basins have a fixed geometry. In the present model, the tributary glaciers are just considered to be buckets. When they have a positive budget they spill over and supply mass to the main glacier. The model has no intention to describ the physical process of how the tributaries flow into the main stream. Although in reality a tributary glacier with a negative mass budget can for a short period still deliver some mass, this will always be a small amount during a limited period of time.*

e) Prescribing surges prevents the model to be applied for predictions (although the conclusion that the surge occurrence does not have long-lasting effects on the glacier front position makes this not so relevant – but at least should be cited).

*We do not understand this comment; for us it is the other way around...*

f) A constant calving rate does not seem a suitable choice for a frequently surging glaciers, as surges imply more intense frontal crevassing and therefore more frequent calving events.

*We will change the calving law to the water-depth dependent version used in earlier applications of the model. At the same time, we think that the impact of smaller bed irregularities (order of 10 m) along a profile which is supposed to represent a width-averaged geometry is limited.*

Many of the model assumptions leading to the above limitations could be maintained in the paper, but they should be discussed as described so the readers are fully aware of the limitations inherent to the model and its results.

**SPECIFIC COMMENTS:**

ABSTRACT:

- P1, L16-18: As the ELA history is reconstructed by matching observed and simulated glacier length because there was a modest correlation between Tunabreen's ELA and Longyearbyen's temperature, I would suggest altering the order of the two sentences in lines 16-18. *Yes*

- P1, L19: As this would be expected due to the bias mentioned in general comment 3, I would try to change the sentence so it does not seem a substantial finding. *Yes*

- P1, L23: becomes -> would become? *Yes*

**1) INTRODUCTION:**

A main modification is this section is to describe Tunabreen and its surges in the context of Svalbard glaciers and surges, as described in general comment 1.

- P2, Figure 2: Perhpas it would be worth noting somewhere that the front positions before the surges (for the first two surges around 25.7-25.8 km; for the last two surges, around 24.7-24.8 km) all are located in the bed bump that can be appreciated at 24- 26 km in Fig. 4 (this is only relevant to observations, not to the modelling results, because of the assumption of a flat submarine bed). *We think the uncertainty in the width-averaged depth is too large to put emphasis on it*

- P3, L53: "effect of reverse bed slopes" and "variable calving rates" are both cited here among the feedback mechanisms that can be dealt with by using MGMs. However, these two are not considered in the present paper. The sentence is correct, as it is general for MGMs, but could induce the reader to think that all of these feedback mechanisms will be considered in the paper. Try to think in a writing that avoids this (or perhaps just say that these two particular ones will not be considered in the present paper).

  *This comment becomes less relevant because we are going to use a water-depth dependent calving law, implying that the effect of various bed slopes is in principal included.*

**2) GLACIER MODEL:**

• P4, L78, eq. (1): perhaps a reader not fully familiar with the mass balance terminology and units could suspect an inconsistency in the dimensions of the various terms in the equation (time derivatives of volume together with [apparent] masses). This could be avoided by describing the units of the various variables involved. Also, in line 79 it would be good to refer to $F$ as *volumetric* calving flux (to distinguish from mass flux) and stating that it is expressed as an specific value over the glacier area (i.e., volumetric calving flux divided by the glacier area). $M_s$ *should be M. Otherwise, we state clearly that mass and volume are equivalent because ice density is taken constant. We will add 'volumetric' at a few places.*

**2.1 *Prognostic equations for glacier length*: 3**

- P4, L89: Some text is missing here (likely, definitions of $N, L, H_m$ and "Deriving"). *The quantities L and W are defined already (line 71).*

- P4, L93, eq. (3): This is eq. (4.2.1) in Oerlemans (2011). Although Oerlemans (2011) is easily downloadable through the Internet, it would be convenient to add here a couple of lines justifying/explaining this parameterization. *The confusion is caused by the fact that in line 103 "of Eq. (2) into Eq. (3) " should be of Eq. (3) into Eq. (2)"*

- P4, L98: Add a reference here supporting this statement. *Yes*

- P5, L111, eq. (6): State that the variable $h$ is surface elevation. *Yes*

**2.2 *Geometry*:**

Important regarding this section are the aspects discussed in general comment 2.

• P6, L144: Similarly, limitations regarding the fixed surface geometry of the tributaries discussed in general comment 4d should be mentioned here. *Yes*

**2.3 *Calving rate*:**

- P6, L150-153: I suggest changing "the dominant control" by "a dominant control", as other controls have been shown to exert an important role (e.g. meltwater filling the crevasses – or enlarging them when flowing downwards through crevasses –, effect of ice mélange), and add to the references listed De Andrés et al. (2018, 2021), doi: 10.1017/jog.2018.61 and 10.1017/jog.2021.27, respectively. *The text will be fully changed since a different calving law will be used (water-depth dependent)*

- P6, L155-156: limitations regarding the constant calving rate discussed in general comment 4f should be mentioned here (with greater detail than that used there). *Yes*

- P6, L162: some text is missing here.

2.4 *Imposing surges*:
Limitations regarding the prescription of surges discussed in general comment 4e should be mentioned here. *Not clear to us; we clearly state what we do*.

2.5 *Climate forcing*:
The bias discussed in general comment 3 should be discussed in this section.
- P8, L202: it was decided -> we decided *Yes, we will*
- P8, L202-203: the sentence "in line with the statement in the beginning of this section" is ambiguous (I imagine that it refers to that on lines 192-193, but the reader should understand it clearly). *Indeed*

3) BASIC EXPERIMENTS ON THE SENSITIVITY OF TUNABREEN TO CLIMATE CHANGE:
- P8, L226 and P9, Figure 5: It would be convenient to comment the case ΔE=+100 m in relation with Fig. 5 and the fact that, in this case, instead of a smooth curve (as in other cases shown in the figure) there is a sharp change some earlier than 1700 yr. We will consider this. It may look somewhat different with the water-depth dependent calving law.
- P8, L232: shows -> show *Yes*
- P9, Figure 6: It would be better to restrict Regime I to the section starting in year 1500 (and ending close to year 3000); Before year 1500 it dos not make much sense. *OK*
- P10, L249-257: The comments in regimes I and II regarding water depth are meaningless under the assumption that the model is using a simplified geometry with flat bed in its submerged part (see general comment 2).

4) SIMULATING THE EVOLUTION OF TUNABREEN DURING THE PAST 100 YEARS:
- P10, L260: Some text is missing here.
- P10, L263-275: The 2002-2004 surge is the only one not discussed.
- P10, L273: I guess that "large part" should be "last one". Additionally, is there an explanation for the statement in the sentence? *Yes, will change to to "last one". There is no obvious explanation why the surge came so fast.*

5) THE FUTURE EVOLUTION OF TUNABREEN:
- P13, L313-314: "a glacier lenght of 16.2 km and no calving anymore" is stated here, but in line 136 $L_1$=17250 m is mentioned. *There is no conflict here because e 16.2 < 17.25*
- P13, L316: I suspect that the last $m^{-1}$ (at the end of the line) is a typo. *Yes*
- P13, Figure 10: Two surges have been imposed, starting in 2030 and 2065. Explain how these values were selected in the light of the dating/frequencies of the previous history of (observed) surges. *Well, it is just a possibility (ambiguity); nobody knows what will happen with the future surging regime*
- P14, L341: Again, the mention that "the front comes into shallow water" is meaningless in the context of a model with simplified geometry with constant thickness of submerged part (i.e., "shallow water" – versus deeper water – does not exist in the model). *Part of the inclined linear profile is still under sea level, so here water depth is variable*

6) DISCUSSION:

The main point here would be the need to summarize the main limitations of the model (as discussed earlier) and how the model results should be taken with caution.

• P15, L359: Add reference to Oerlemans at the end of the first sentence (although mentioned earlier – line 355 –, it will make it clearer). *Yes*

---

## Author Comment (AC2)

**NOTE: Quotations from review in normal face. Our response in *italic***

Authors employ the Minimal Glacier Model (MGM) to investigate the past and future evolution of a surging glacier in Svalbard, Tunabreen. MGM is a simple, analytical model that is based on the principle of mass conservation, where the glacier length and thickness change are related to the net mass balance (mass exchange with atmosphere and ocean).

*Note: the model is not 'analytical' in the usual sense. The glacier system is reduced to a zero-dimensional nonlinear differential equation that is solved numerically.*

Previously, it has been successfully applied by the leading author to other glaciers in Svalbard (Hansbreen, Abrahamsbreen, Monacobreen) and despite (or, in fact, thanks to...) its simplicity gave insights to the processes governing their long-term changes. After reading the manuscript I was left with a question about the novelty of the results compared to previous work and I hope that authors can clarify this in the revised manuscript.

*The fact that a certain type of model is applied many times does not reduce the 'novelty' of a study, as the purpose of our paper is not to present a new model. Other models (i.e. SIA) have been applied to many glaciers. We note that Tunabreen has never been modelled! In fact, the number of studies in which SIA or higher-order models with proper calibration over the past have 100 years have been applied to Svalbard's tidewater/surging glaciers is close to zero. The question we approach is that of the future of Svalbard glaciers – a very valid question in our view. As we write in the discussion "Although in the end one would like to repeat the simulations presented here with a comprehensive glacier flow model with spatial resolution, it will not be an easy task to prepare the necessary input fields and get the calving and surging at the right place and time!" So we believe our study is original and relevant.*

In the current implementation of the model, surging is prescribed as a change in ice thickness that can be interpreted as a change in basal conditions that causes enhanced sliding at the bed. Owing to the model structure, several assumptions and simplifications need to be made in order to tune the model to observed record of glacier length. Many of those assumptions are reasonable, nonetheless some are not that easy to justify. My main concern is the choice of a constant calving rate and flat bedrock in the ablation zone.

*We will change the calving law to the water depth-dependent version used in earlier studies and calibrate it with the measurement.*

In my opinion, one of the important deficiencies of this study is a lack of external validation of the model, as it is tuned to entire data set of glacier length. Where there is a possibility to confront the model performance with an independent data set - runs with $ELA_{LYR}$ as the meteorological forcing - the results are not satisfactory and authors prefer to revert to a synthetic climatic forcing based on the inverse modelling approach (which may be considered as a possible over-fitting of the model).

*The uncertainty in the ELA record from meteorological measurements/modelling is quite large and it is not possible to decide whether the discrepancy between observed and simulated glacier length is due to a model deficiency or to poor quality of the forcing function. A good validation of the model would only be possible were there any ELA (i.e. mass balance) measurements on Tunabreen. But as we all know, this is not the case.*

*We do not quite understand what is meant by 'overfitting'. Our philosophy simply is that a projection of future behaviour should be based on a proper simulation of the past record. Otherwise, the initial state (imbalance) is not correct and may cause an unrealistic evolution of the glacier.*

Clearly, one of the plausible explanations of discrepancies in the simulated glacier length may be the mass imbalance caused by under- or overestimation of the frontal ablation, mainly due to the use of a constant calving rate. As shown by the authors, the model is indeed highly sensitive to the choice of calving rate (e.g. Figure 5) and therefore any error in the estimation of this variable can have a profound impact on the final results. I do agree that calving rates do  not necessarily strictly follow a surge cycle - see Mansell et al. (2012) who confirm rather modest changes in calving on some surging glaciers.  However, if we look at the frontal ablation of Kronebreen for example, we can observe a variability between consecutive years reaching 50% (e.g. Kohler et al., 2016). I do agree that a robust modelling of calving rates of Tunabreen in a longer time perspective (100 years) may be beyond our capacity and it may be easier to stick to one value as authors have chosen. I would like to see a convincing explanation why did authors decide to completely disregard calving rate parametrizations based on the water depth criterion (e.g. Mercenier et al., 2018). An argument that calving depends mostly on water temperature is based on studies covering relatively short period (e.g. Luckman et al. (2015) study covered only a 1.5 year) and therefore cannot be considered reliable over longer timescales. On the other hand, authors have previously applied in MGM parametrizations of calving rate based on the water depth criterion (e.g. Oerlemans et al., 2011) and the results were convincing. I wonder why wouldn't it work for Tunabreen as well? In a comment to the Figure 6 the authors stress how important the glacier geometry is for response to climatic forcing, especially the location of a point where it switches from tidewater to land based. Yet they disregard fluctuations of the water depth along the longitudinal profile by assuming presence of a flat bed down-glacier from point $L_t$.

*As noted  above, we will change the calving law to the water depth dependent version used in earlier applications of the model. At the same time we think that the impact of smaller irregularities of the bed (order of 10 m) along a profile which is supposed to represent a width-averaged geometry should not be overestimated.*

**Specific comments:**

line 60, Figure 3: Maybe include the sea floor bathymetry as well
*The sea floor bathymetry is relatively flat in the inner fjord, and while we can add a contour line or two, it really does not add that much to the figure.*

lines 73-74: This set up of x-axis assumes a stable position of the ice divide which, generally, doesn't have to be met as there are documented examples of the ice divide migration during a surge (e.g. Fridtjovbreen)
*This is true. A model without spatial resolution cannot handle this. However, we believe that in the case of Tunabreen the effect of a changing ice divide on the overall mass budget is limited. But this is hard to demonstrate...*

line 79: In Eq (1) there is $M_m$, not M. Shouldn't F be the frontal ablation?
*Indeed, $M_m$ should not have the index. We are not quite sure what is meant by 'frontal ablation'. The calving flux is the ablation rate times the ice thickness at the front times the glacier width.*

line 94: If you use bars as notation of a mean, why not use H-bar as well instead of $H_m$?
*Yes, for consistency that would have been better. However, in earlier model descriptions (notably Oerlemans, 2011), $H_m$ was preferred because it is easier to use as a label in graphics. We will change this.*

line 103: Is is time derivative of Eq. (2) that is being substituted to Eq. (3) or is it the opposite? For more clarity, at least some intermediate steps of this substitution should be provided. Is Eq. (4) complete?

*It is the opposite, which causes the confusion. The derivation can be found in Oerlemans (2011), which can directly be downloaded from the internet.*

line 110: b-dot is a balance rate while b-bar is the mean bed elevation, it is confusing to use the same letter b for two distinct variables

*We do not see the problem. A bar or dot has the same function as an index.*

lines 119-122, Figure 4: there is only one longitudinal radar profile between km 12.5 and 20 (Figure 3), how was the band average of bed topography calculated over this section?

*At L124, we state that the bed topography is based on the Fürst et al (2018) reconstruction as well as the 2015 helicopter radar; we will also add that non-glaciated surface topography (i.e. the mountains) is used to constrain the bed reconstruction.*

The bed topography between 0 and 14 km looks like it would have been approximated much better with a parabola than a straight line.

*It would lead to a much more complicated model formulations, but in the end it would have a limited impact as long as $L > 15$ km, because it is the mean bed profile that matters.*

lines 129-130: How about the undulations in the lower part of the glacier? Would they have a limited impact as well?
lines 135-136: Yet in Figure 3 one can clearly see that the bed goes below the sea level at km 13-14

*The bed undulations in the lower part of the glacier have an amplitude of typically 10 to 20 m, which is comparable to lateral (across-fjord) variations as seen in the bathymetry in front of the glacier (see map at https://toposvalbard.npolar.no; last accessed 2 September 2021). In view of the flowband geometry adapted here, the undulations are thus seen as irregularities with only a local effect on the calving process. We also note that due to depositional and erosional processes the glacier bed may be subject to significant changes even within a time span of a hundred years.*

lines 133-141: Please be consistent with the order of subequations a and b, once $x<L_i$ is placed first, later $L>L_i$.

*L is not x. Together with the description after Eq. (8b) it is clear.*

line 140: Either close the bracket or remove it
*Yes*

lines 146-148: How do you explain the physical meaning of this decoupling? Ice should flow from the tributaries to the main trunk regardless of their net mass balance as long as they are dynamically connected.

*In the present model, the tributary glaciers are just considered to be buckets. When they have a positive budget they spill over and supply mass to the main glacier. The model has no intention to describe the physical process of how the tributaries flow into the main stream. Although in reality a tributary glacier with a negative mass budget can for a short period still deliver still some mass, this will always be a small amount during a limited period of time. We will add a few sentences to make it clearer.*

line 156: Can you provide some uncertainty estimation of the assumed calving rate, for example its standard deviation over 2012-2019?

line 158: $\tan^{-1}(d/3)$ is equal to ~1.5 for d=40 m. I hope this is just a typographic error in Eq. (12). Otherwise there is a problem with mass conservation in the model as F is overestimated by roughly 50% when $L>L_i$.

*The calving law can easily be changed and all the calculations can be redone.*

line 165: d is missing in the second term in the bracket.

*Will be corrected*

lines 166-168: I wonder how often is the second criterion met in your simulations, especially when $H_m$ decreases during a surge? Shouldn't calving rate increase substantially when the glacier reaches flotation?

*The criterions is always fulfilled, as the reduction of the frontal thickness is not so large (max 10 m for the largest surge). It is perhaps a bit counter-intuitive, but a survey of the literature shows that there is no systematic relation between the phase of a surge and the calving rate. We will add references.*

lines 200-202: Have you considered adding some noise to this smooth function, possibly with the same variance as observations?

*This makes little change since the response time is long and fluctuations are immediately integrated.*

lines 209-210: How does this synthetic ELA record compare to ELA calculated with Van Pelt et al. (2019) model for years 1957-2020 (lines 194-196)?

*The main difference is the larger trend in Van Pelt et al. (2019) [ELA-rise of 4.6 m per year, as compared to 2.5 m per year from the reconstructed forcing]. We will mention this.*

lines 252-253: Calving rate in your study is constant. Second, I wonder if this regime cannot be explained with your assumed $\tan^{-1}(d/3)$ term in Eq. 13 that can be considered more as a trick to make the calving flux go down to 0 smoothly at the 0 water depth as mentioned in lines 158-159.

*With the new calving law this will be reconsidered.*

lines 259-260: How sensitive is the model to the choice of these parameters? Can you provide more details on the optimization procedure you applied and its results?

*The control parameters have just been varied to provide the best fit between observations and simulation in a rms-sense. The behaviour with respect to the control parameters is smooth, and to present an extensive analysis really serves no point in our judgement. In fact, the degree of sensitivity is demonstrated by the experiment depicted in Figure 8.*

lines 256-257: Yet the assumption of a flat bed in the ablation zone (Eq. 8b, $b(x)=b_d$ when $x>L_1$) disregards such feedbacks during frontal recession, whereas numerous studies have shown that calving rates decrease when glacier reaches a pinning point or shallow bed

*'Numerous studies have shown'? Perhaps a few qualitative analyses of some extreme cases (like Columbia glacier). However, we will formulate this point more carefully...*

lines 265-266: Can you compare your modelled mass balance perturbations due to a surge with any observations, either from Tunabreen or some other Svalbard glacier?

*To our knowledge there are no mass balance programs on surging glaciers in Svalbard, except for Kongsvegen (but not before the last surge in 1948).*

Figures 7 and 8: Having same axis on both figures would make them easier to compare. Km 23 is missing in Figure 8 y-axis.

*KM 23 is not missing – this is on purpose (like the scale for E on the right-hand side). We tried to get the maximum of information in the figures in a transparent way – apparently not appreciated....*

lines 295-299: Are there any other plausible explanations of this discrepancy? How about changing calving rate, or more generally, assumed simplifications in the frontal ablation calculations in the model?

*Yes, varying calving rates is another candidate. We will discuss this in more detail. We add at the end of section 4:*

*The mass-balance simulation with a regional climate model by Van Pelt et al. (2019) yields a mean increase of 4.6 m a$^{-1}$ of the ELA over the period 1957-2018. This is substantially larger than the 2.5 m a$^{-1}$ found for the reconstructed forcing. The discrepancy is substantial and hard to explain. However, since the simulation by Van Pelt et al. (2019) does not go further back in time than 1957, a thorough comparison remains difficult.*

*Referring to the sensitivity of the glacier length to the calving parameter (Figure 5), it is obvious that the use of a fixed calving parameter is a limitation of the calibration procedure. It would even be possible to keep the ELA fixed and vary c. Possibly variations in c and the ELA work in parallel, because both quantities presumably increase with atmospheric temperature (assuming that water temperature in summer is related to air temperature). However, increasing c would imply a smaller increase in the ELA, and the discrepancy between the reconstructed ELA history and the simulation by Van Pelt et al. (2019) would become larger.*

lines 313-314: According to Figures 3 and 4, calving would stop further inland, at km 13-14.

*The relevant line in Figure 4 is the gray line 'Model b', where the bed goes below sea level at ≈17 km.*

Figure 10: In the Paris run, there appears to be a sufficient time lag between the surges to mimimize their effect on the glacier length change, contrary to the recent two observed surges (as described in lines 274-275). How would this prognostic simulation change if there was no time for the glacier to adjust between the surges?

*A few calculations will be done on this. However, the number of possible ways to describe the surge frequency / amplitudes is endless....*

lines 370-371: This conclusion was reached with the same approach/method as used in this paper. How about other studies, do they confirm your findings about the small impact of surging on the long-term evolution of the glacier length?

*Yes, it would be so nice if other studies with different models (SIA, higher-order) would be done to investigate this point. But to our knowledge, no such study has been carried out....*

---

## Author Response (AR1)

Dear Dr. Martin, Dear Reviewers,

We are pleased to submit a revised manuscript in which we have dealth with the numerous comments and suggestions of the reviewers as *indicated in our initial responses.*

As most notable points we mention:

• The limitations of the MGM have been spelled out in more detail.

• The previous calving law has been replaced by a water-depth dependent calving law, and *all the calculations have been redone*. The results differ not so much from the previous calculations.

• The discussion on the climate forcing has been extended, and it is explained why the simulation of Van Pelt et al, when used as forcing to the glacier model, does not give a good result.

• Figures 5 - 11 have been replaced by updated versions.

We hope that the reviewers appreciate all the changes we made and will be positive about the result.

With kind regards, also on behalve of the co-authors,
Johannes Oerlemans

Utrecht, December 21, 2021

---

## Referee Report (RR1)

Paper No.: tc-2021-155

Author(s): Johannes Oerlemans, Jack Kohler, Adrian Luckman

Title: Modelling the mass budget and future evolution of Tunabreen, central Spitsbergen

Referee: Francisco Navarro

**GENERAL COMMENTS:**

I acknowledge the efforts by the authors to address and clarify the main points suggested by the reviewers, which have led to a substantially improved version. Particular clarifications such as the one regarding the consideration of the tributary glaciers as buckets, which, when have a positive budget spill over and supply mass to the main glacier, are specially useful. However, in my view some aspects are not yet sufficiently discussed/stressed, justified or taken into account. I will focus on these, as general comments, because the paper, after its first review and subsequent revision, virtually does not need further specific comments. My comments follow a sequential order which does not imply any relative relevance.

1) CONTEXT: In my first review I asked the authors to add some comments to put the paper into context. I suggested them to address the following questions:
   – Is it a large/medium/small tidewater glacier compared to the rest of Svalbard tidewater glacier?
   – Is it surface slope within the usual range?
   – Are its typical calving rates similar to those typical of other Svalbard glaciers (or higher/lower than usual)?
   – How many known surging glaciers have been identified in Svalbard?
   – What is the usual range of surging periods?
   – Do usually glacier surges in Svalbard initiate at the front and propagate upwards?
   – Has any other Svalbard glacier known to have experienced surges with increasing frequency?

   The answer from the authors has been "*Our paper is not intended to be a general paper about surging, but is instead focused on providing a model of a glacier that has undergone some surges. Here we face the usual conflict between a journal that wants a paper to be compact and concise, and a reviewer that wants to see more extensive descriptions. However, we will add a few lines to provide more context.*"

   I was certainly not asking for an extensive description of the above-mentioned points but just a paragraph helping the readers not so familiar with Svalbard glaciers to understand the context of this study and its relevance. This should be not difficult for the authors, given their expertise on Svalbard glaciology. In any case, I have not found in the introduction those "*few lines to provide more context*".

2) SIMPLIFIED BED GEOMETRY AND CALVING LAW: In the previous review the implications of the simplified bed geometry with a flat submerged part (lack of effect on the advance/retreat rates, e.g. fast retreat on reverse-bed slopes, and stability conditions) were pointed out. It was also pointed out that a constant calving rate does not seem a suitable choice for a frequently surging glacier, as surges imply more intense frontal crevassing and therefore more frequent calving events. The authors

have addressed this point mainly by changing to a water-depth dependent calving law of the form $F = -c\, d\, W H_f$. It is clear, however, that a water-dependent calving law is not a suitable choice (in the sense that this particular property adds nothing) for a (simplified) flat submerged bed, as this implies a constant water depth. The change from a constant calving rate to a calving law as described, involves, however, a certain improvement, because the new law reflects the changing (in time) glacier cross-sectional area, which makes the calving flux nonconstant. But, as noted, the authors should not emphasize the "water-dependent" characteristic of the calving law, which has no practical effect given the flat submerged geometry.

Regarding the possible dynamical effects of the simplification of the submerged bed geometry (to a flat bed), the authors have added (among others) in the new version of the manuscript the sentence *"The bed undulations in the lower part of the glacier have an amplitude of typically 10 to 20 m, which is comparable to lateral (across-fjord) variations as seen in the bathymetry in front of the glacier… In view of the flowband geometry adapted here, the undulations are thus seen as irregularities with only a local effect on the calving process."* However, bed undulations of that size do not only have an a local effect on the calving rates, but can also have an influence on the stabilization of the glacier front position changes. Although numerical models used to illustrate such stabilizing effect usually employ larger undulation amplitudes (e.g. 50-100 m in Vieli et al. (2001)), recent stable front positions of glaciers such as Hansbreen (another Svalbard tidewater glacier) involve much lower amplitudes, of the order of those of Tunabreen (see e.g. Figure 2 of Otero et al. (2017), to get a sense of the scale of such undulations).

3) MODEL LIMITATIONS: After reading the arguments by the authors in their answer to the comments in my first review, I concur with them in that their tuning of the ELA history with the glacier length observations is the only thing that they could do given the available data. As the authors admit that their former statement *"the simulated glacier retreat is in good agreement with observations"* is *"a bit silly"*, I have also to admit that my own statement *"which makes the model results of limited value"* was too strong. My intention was to remark that the mentioned agreement is to be expected, as it has been used for the tuning. But, obtained such an agreement, the model can (of course, with limitations) be applied in prognostic mode as done later in the paper.

I am also sorry about the misunderstanding regarding my comment "Prescribing surges prevents the model to be applied for predictions (although the conclusion that the surge occurrence does not have long-lasting effects on the glacier front position makes this not so relevant)". I should have clarified that the first part of the sentence applied to short-term predictions of front position (in particular, to prevent surges - this is crystal-clear). But the second part of the sentence makes clear that, given that the observed surges did not have long-term effects on the glacier front position, the model could be used for prognostic purposes, as done in the paper (once again, with limitations). So I basically agree in this with the authors.

Lastly, I should emphasize that I acknowledge the use of simple models (such as the MGM used here) to analyze the interactions between glaciers and climate, especially for long-term simulations, and to analyze the sensitivity to the model parameters. But special care has to be put into the selection of the simplifications and the analysis of their possible effects. In other words, in not trying to reach not sufficiently sustained

conclusions and in making always clear the model limitations (and this was the aim of my comments regarding the model limitations).

REFERENCES:

Otero, J., Navarro, F.J., Lapazaran, J.J., Welty, E., Puczko, D. & Finkelnburg, R. (2017). Modeling the Controls on the Front Position of a Tidewater Glacier in Svalbard. Front. Earth Sci., 5, 29, doi:10.3389/feart.2017.00029.

Vieli, A., Funk, M., & Blatter, H. (2001). Flow dynamics of tidewater glaciers: A numerical modelling approach. Journal of Glaciology, 47(159), 595-606, doi:10.3189/172756501781831747.

---

## Author Response (AR2)

**1. To meet criticism by Navarro about 'perspective' we have inserted at line 35 (+ references in the reference list):**

Svalbard has a wide spectrum of surging glaciers, with surge periods ranging from a couple of decades to (presumably) a few hundred years (Lefauconnier and Hagen, 1991). Many of the larger tidewater glaciers on Svalbard do surge regularly (a precise percentage is not known). Glaciers with well-documented surge behaviour in a similar size-class as Tunabreen are for instance Kongsvegen (length $\approx$ 25 km, slope $\approx$ 0.028), Comfortlessbreen (length $\approx$ 16 km, slope 0.042) and Monacobreen (length $\approx$ 40 km, slope $\approx$ 0.027). The characteristic slope of Tunabreen ($\approx$ 0.032) is within the usual range of larger tidewater glaciers. The observed long-term (multi-annual) calving rate is relatively high ($\approx$ 270 m a$^{-1}$), but it should be noted that calving rates vary enormously across the Archipelago (Blaszczyk et al., 2009).  The surges of Tunabreen appear to be initiated on the lowest part of the glacier and propagate upward, which is not uncommon for surging tidewater glaciers in Svalbard (Sevestre et al., 2018). Altogether, Tunabreen appears to be a fairly representative glacier, albeit with a relatively high and increasing surge frequency.

Błaszczyk, M., Jania, J., Hagen, J.O.: Tidewater Glaciers of Svalbard: Recent changes and estimates of calving fluxes, Polish Polar Res., 30(2), 85-142, 2009.

Lefauconnier, B., and Hagen, J.O.: 1991. Surging and calving glaciers in eastern Svalbard. Nor. Polarinst. Medd. 116, 1991.

Sevestre, H., Benn D.I., Luckman, A., Nuth, C., Kohler, J. Lindbäck, K. and Pettersson, R.: Tidewater glacier surges initiated at the terminus, JGR Earth Surface, 1035-1051, doi: 10.1029/2017JF004358, 2018.

**2. Bed undulations (point by Navarro)**

This remains a difficult issue. We acknowledge that bed undulations may facilitate stable positions of the front. However, in our view it remains questionable if considering small undulations in only one-dimension makes much sense. The reviewer refers to the study by Vieli et al. (2001), where a numerical calculation is done along a flowline. But, when lateral undulations are important, the choice of flowline becomes critical as well, and perhaps a 2-D approach is necessary to fully capture the effect of undulations..

We believe that in our case bed undulations are less dominant for the dynamic evolution because we consider longer time scales and because Tunabreen is subject to vigorous surges during which the front overrides undulations anyway (and which may actually *change* the bed).

**3. Significance of the work (Anon. Reviewer, AR)**

We regret that AR classifies the impact of this article as "fair".

There is a great need to quantify how the larger glaciers of Svalbard will behave in the future. Model studies undertaken so far are of a very global nature (in effect only looking at mass balance, e.g. Hock et al., 2019), not dealing with the pecularities of calving and surging processes at all.  By no means do we claim to have the perfect model, but we deal with the interaction between dynamics and mass balance, which is a step forward. In the discussion we write: "*Therefore, in the end one would like to repeat the simulations presented here with a comprehensive glacier flow model with full spatial resolution. However, it will not be an easy task to prepare the necessary input fields, formulate boundary conditions in a straightforward way, and get the calving and surging to occur at the right place and time*".

Hock et al.: Glacier MIP: A model intercomparison of global-scale glacier mass-balance models and projections. Journal of Glaciology , Volume 65 , Issue 251 , June 2019 , pp. 453 – 467, DOI: https://doi.org/10.1017/jog.2019.22

**4. Geodetic mass balance (AR)**

Meanwhile to DEMS have been published for Svalbard, en this has allowed us to make a comparison between our model simulation calibrated on glacier length and the geodetic evidence.

**We thus have inserted at original line 311:**

Two Digital Elevation Models (DEMs) for Svalbard have recently been published, referring to the years 1936 and 2010 (Geyman et al., 2022). The DEMs allow an independent check on the model performance. The mean change in surface elevation for Tunabreen over the period 1936 – 2010, based on gridded DEM data,  amounts to $-0.30$ m w.e. a$^{-1}$. The mean net balance calculated for this period from the model output (run of Fig. 7) is $-0.25$ m w.e. a$^{-1}$. We thus conclude that the calibrated model result is in broad agreement with the geodetic evidence.

Geyman, E.C., Van Pelt, W. J. J., Maloof, A. C., Faste Aas, H. and Kohler, J.: Historical glacier change on Svalbard predicts doubling of mass loss by 2100, Nature, 601**,** 374–379 (2022). https://doi.org/10.1038/s41586-021-04314-4

**5. Calculated ELA trend (AR)**

We have access to all ouput data from the study of Van Pelt et al. (2019), so we assume that our calculated value of the ELA trend, derived directly from the model output, is more accurate than the back-of-the-envelope estimate by AR.